# Assessing the Environmental Performances of Nature-Based Solutions Implementation in Urban Environments through Visible and Near-Infrared Spectroscopy: A Combined Approach of Proximal and Remote Sensing for Monitoring and Evaluation

**Giuseppe Bonifazi** [1,2,*] **, Riccardo Gasbarrone** [3] **and Silvia Serranti** [1,2]

1    Department of Chemical Engineering, Materials and Environment, Sapienza-University of Rome, Via Eudossiana 18, 00184 Rome, Italy; silvia.serranti@uniroma1.it
2    Research Center for Biophotonics, Sapienza-University of Rome, Polo Pontino, Corso della Repubblica 79, 04100 Latina, Italy
3    Research and Service Center for Sustainable Technological Innovation (Ce.R.S.I.Te.S.), Sapienza-University of Rome, 04100 Latina, Italy; riccardo.gasbarrone@uniroma1.it
*    Correspondence: giuseppe.bonifazi@uniroma1.it

**Abstract:** The implementation of Nature-based Solutions (NbS) in urban environments is gaining momentum as a means to address environmental challenges and promote sustainable development. However, effective monitoring and evaluation are essential to assess the performance of NbS interventions and to guide decision-making. This research paper introduces a combined approach of proximal and remote sensing, based on visible and near-infrared spectroscopy, to monitor and evaluate NbS implementation in urban areas. The study focuses on the case of the UPPER (Urban Productive Parks for Sustainable Urban Regeneration) project and aims to establish urban Productive Parks as a novel NbS approach in the town of Latina (Italy). Field-based proximal sensing techniques (i.e., near-infrared spectroscopy, NIR) and satellite-based remote sensing data from the Sentinel-2 mission are employed. By integrating these techniques, the study enables comprehensive and multi-scale monitoring of vegetation health and assessment of vegetated areas. Various band ratio indices are calculated to assess vegetation coverage, water content, and urbanization. Temporal variations in these indices are analyzed to evaluate the effectiveness of NbS interventions and their impact on the urban environment. The combined approach of proximal and remote sensing demonstrates the potential for comprehensive and multi-scale monitoring of NbS in urban environments. The research findings contribute to the existing knowledge on NbS monitoring and evaluation, providing valuable insights for sustainable urban development and evidence-based decision-making.

**Keywords:** nature-based solutions; monitoring and evaluation; near-infrared spectroscopy; NIR; NbS; Sentinel-2; remote sensing; proximal sensing

## 1. Introduction

### 1.1. Nature-Based Solutions Overview

Over the past decade, the concept of Nature-based Solutions (NbS) has gained prominence in the fields of environmental sciences and nature conservation [1]. It has been adopted by international organizations seeking effective approaches to address the impacts of climate change, promote sustainable livelihoods, and safeguard natural ecosystems and biodiversity. The emergence of NbS reflects a growing recognition of the need to collaborate with ecosystems to adapt to and mitigate the effects of climate change, while simultaneously enhancing sustainability and preserving the integrity of natural environments. In this context, NbS aim to achieve society's development objectives while protecting human well-being [2].

The European Commission (EC) provided the first official definition of NbS as interventions that effectively address environmental, social, and economic challenges in a synergistic manner [3]. Another important definition of NbS was provided by the International Union for Conservation of Nature (IUCN). The IUCN defined NbS as actions/initiatives focused on preserving, regulating, and regenerating natural or altered ecosystems [4]. The idea of NbS is recognized, in the most recent report of the EC on the topic [5], as an innovative approach to socio-ecological adaptation and resilience.

Despite the promotion of the use of NbS as a critical strategy for addressing numerous environmental and societal problems, the idea and its real-world implementations are not entirely clear [6]. This uncertainty is caused by the NbS concept's evolution from the synthesis of numerous scientific disciplines. In order to provide a clearer understanding of the NbS concept, the IUCN Global Standard comprises a set of criteria that serve as a framework for categorizing blue-green interventions (i.e., interventions in marine areas—blue, or on land, both in rural and urban settings—green [7]) as NbS actions [1,4,8]. The IUCN's set of principles provides guidance for the development and implementation of NbS, ensuring their alignment with sustainability goals and the holistic management of environmental and societal issues [2]. NbS have been also defined as solutions that utilize natural capital to bring benefits to both ecosystems and the human communities reliant on them [9]. The result is that NbS produce a wide range of services that can be categorized into three broad groups: social, environmental, and economic [6]. These solutions may be employed independently or in conjunction with other approaches to societal challenges. Eggermont et al. (2015) proposed a typology that characterizes NbS interventions according to two criteria: (i) the level of engineering required for ecosystems and biodiversity involved in the NbS, and (ii) the extent to which the NbS can enhance ecosystem services [10].

In the context of urban areas, NbS rely on utilizing or creating natural features and elements within and around cities to provide essential ecosystem services [11]. These services, in turn, contribute to various aspects of sustainable urbanization. NbS have the potential to address multiple interconnected challenges at different scales [12]. These challenges include (i) fostering economic development in urban areas [13], (ii) enhancing the social aspects of sustainable urbanization [14], and (iii) generating positive environmental impacts [10]. In terms of environmental benefits in urban areas due to NbS, these encompass the regulation of climate, air quality, water treatment and purification, natural hazards and waste management, erosion control, green space management, pollination, and more [3,15]. The extensive implementation of NbS in urban areas is expected to have a transformative impact on Urban Planning and Development (UPD), assuming crucial roles in several key areas [12]. One of the primary benefits of NbS integration is its contribution to climate change mitigation and adaptation strategies. By incorporating green/blue infrastructures and natural elements into urban spaces, cities can reduce their carbon footprint, enhance carbon sequestration, and mitigate the urban heat island (UHI) effect [16], ultimately helping to combat the effects of climate change [17]. NbS can significantly enhance urban resilience and risk management. By creating or restoring natural habitats (i.e., wetlands and green spaces), cities can better manage stormwater runoff, reducing the risk of flooding and water-related disasters [18]. Additionally, NbS promote the creation of multifunctional and interconnected green spaces that support biodiversity and ecological balance within urban environments [15]. Moreover, the incorporation of NbS into UPD can yield various co-benefits. For instance, the enhancement of green spaces in cities not only improves air quality and reduces noise pollution but also contributes to the well-being and mental health of urban residents (i.e., nature-rich environments have been linked to reduced stress levels) [14].

*1.2. Monitoring and Evaluation of NbS Implementation*

In each case study, it is crucial to verify the impacts of NbS through monitoring and evaluation processes. This verification not only provides evidence but also informs the

decision-making process, allowing for adaptive management [19]. In order to ensure sustainable and resilient development in the face of climate change, urban planners in both the public and private sectors must establish a baseline and adequately monitor and assess the impacts of NbS interventions [20].

Raymond et al. (2017) proposed a framework for assessing NbS' co-benefits related to their implementation in urban areas [15]. In addition to highlighting the advantages and disadvantages of NbS that result from already-existing ecosystem services [21,22], the framework advances current understanding by also recognizing the advantages and disadvantages of interactions between ecosystems, biodiversity, climate, economic systems, and socio-cultural elements [17]. The monitoring and evaluation of co-benefits is viewed as a cross-cutting aspect that necessitates the involvement of specialists skilled in impact assessment at each level of the process of NbS implementation [15].

In this context, visible and near-infrared (Vis-NIR) spectroscopy can be seen as a valuable tool for monitoring NbS implementation. Such a technique utilizes the absorption/reflection properties of light to analyze the composition and characteristics of materials, including vegetation and soil. Vis-NIR techniques are widely used in the agri-food sector for different purposes [23–26] and for soil analyses [27–29]. In this scenario, the utilization of this sensing technique, both at 'proximal' and 'remote' scale, enables non-destructive and rapid assessment of key parameters related to plant health, nutrient content, and overall ecosystem functioning [30].

Earth Observation (EO) data, acquired from satellites or aerial platforms, provide broad-scale and multi-temporal information on vegetation dynamics, land cover changes, and ecosystem health. This allows for the assessment of NbS' impacts on a larger spatial scale [12]. Proximal sensing, on the other hand, involves the use of handheld or ground-based sensors to collect high-resolution data on specific parameters such as plant stress, biomass, and soil moisture. Proximal sensing enables detailed monitoring at a local scale and facilitates real-time decision-making for NbS management.

Remote and proximal sensing techniques, such as NIR spectroscopy, contribute to NbS management (Figure 1), especially in the monitoring and evaluation stage [12]. Those techniques allow us to verify the environmental positive impacts of NbS. However, such impacts have to be verified through monitoring and evaluation phases from an adaptive management perspective [31].

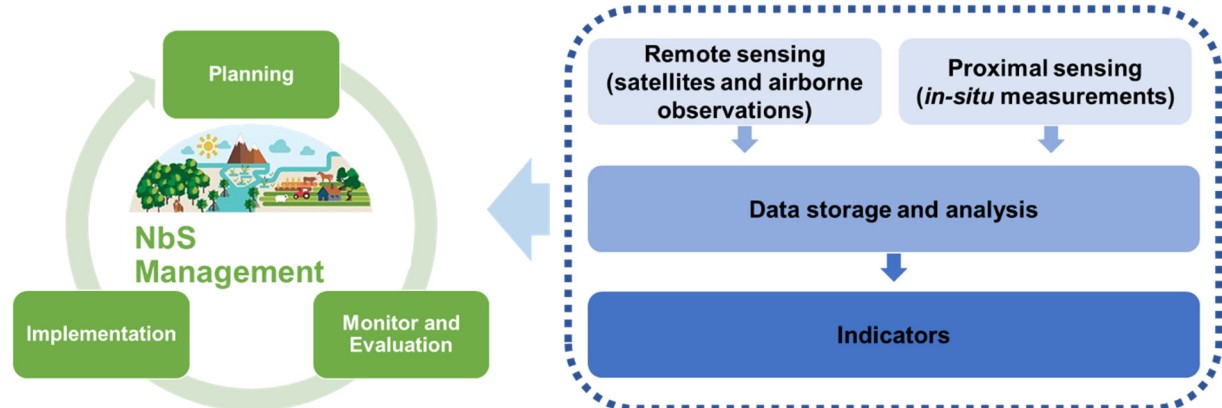

**Figure 1.** Remote and proximal sensing techniques' contribution to the management of Nature-based Solutions (NbS). Proximal and remote sensing data and products allow us to generate indicators that can be used in the NbS monitoring and evaluation stage [12].

*1.3. UPPER Project—Urban Productive Parks for the Development of NbS Related Technologies and Services*

The presented case study is framed in the UPPER project (https://uia-initiative.eu/en/uia-cities/latina (accessed on 10 January 2023)). One of the main objectives of the

UPPER project is the experimentation with urban Productive Parks dedicated to the co-production of NbS to address different social, environmental, and economic challenges in the city of Latina (Latina, Italy). In more detail, the socio-economic challenges included: (i) limited availability of accessible public green spaces per resident, below the national average; (ii) underutilization of existing parks due to inadequate facilities and perceived safety concerns; (iii) a high concentration of unemployed and inactive individuals; (iv) brain drain towards other regions; (v) exploitation of illegal immigration and social exclusion, affecting both minorities and vulnerable populations; (vi) scarcity of public economic resources; (vii) lack of coordination between public and private entities; (viii) dispersion of specialized expertise, posing difficulties in rehabilitating ecosystems and ensuring long-term maintenance of green and blue infrastructure. And the interconnected environmental challenges regard: (ix) rapid and unregulated urbanization leading to extensive use of land and natural resources; (x) detrimental effects on the delicate ecosystem of canals and rivers, including high pollutant concentrations and hydrogeological risks; and, finally, (xi) increasing occurrence of extreme weather events (i.e., heatwaves and sudden floods).

In order to face these challenges, the UPPER project takes a comprehensive approach by integrating greenery, green infrastructure, and innovative outdoor services and activities.

It aims to revitalize underutilized urban areas, creating Demonstration Sites for testing self-produced NbS interventions such as phytoremediation, indigenous tree planting, and soft engineering measures. The Productive Parks will be co-designed and co-managed by project partners, local citizens, and stakeholders, with a focus on long-term maintenance and sustainability. The initiative also explores the Green Areas Bank to meet the demand and supply of NbS, while integrating a job and skills program for vulnerable citizens. Support for social enterprises and for-profit startups will ensure market exploitation and further development of the Productive Parks, ultimately aiming for wider implementation and impact.

In the UPPER project, NbS are, then, intended as actions/measures that are supported by nature and jointly benefit the environment, society, and the economy [9]. The area designated for the NbS interventions within the UPPER project are reported in Figure 2.

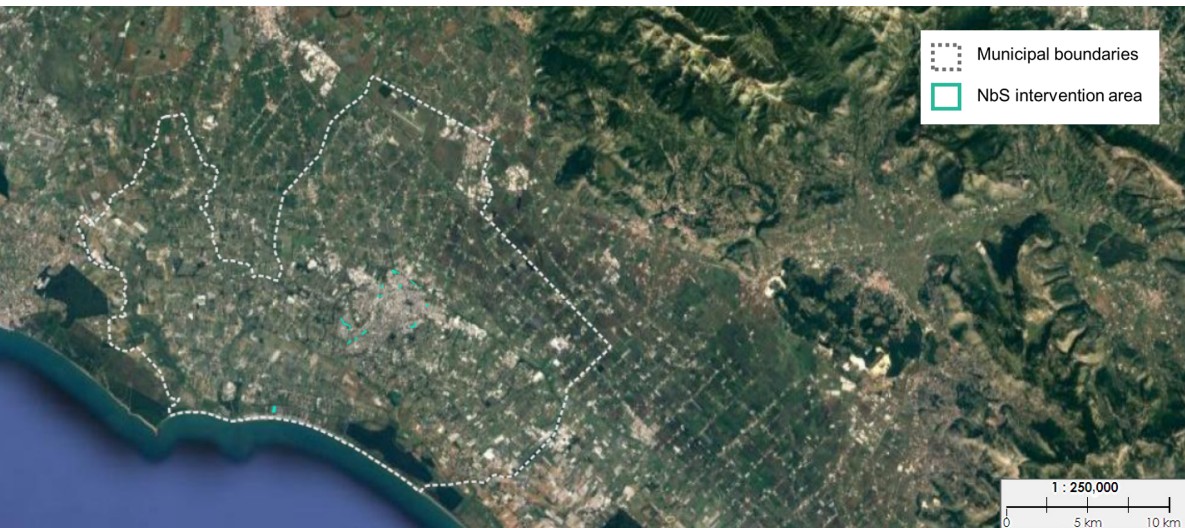

**Figure 2.** Territorial context of the municipality of Latina and the planned NbS intervention areas within the UPPER project (2020). Map was extracted and elaborated within the WebGIS Holding Interactive Platform (WHiP) (https://www.cersites.gter.it/webgis_lpdt/lizmap/www/index.php/view/map/?repository=lab&project=UPPER_WHIP (accessed on 17 November 2022)).

The approach presented in this study focuses on the application of remote sensing and proximal sensing techniques, supported by meteo-climatic data, for the environmental monitoring of NbS interventions within a European project. Specifically, in this study,

environmental monitoring refers to the assessment of changes from the ex-ante phase (or baseline state) to the NbS implementation phase, in terms of plant health and vegetation area evaluation.

The study seeks to provide a comprehensive understanding of the environmental changes resulting from NbS interventions by combining remote sensing data at a broader scale, to evaluate specific indicators at a synoptic level, and point spectroscopy data at a localized and small-scale level. This integrated approach enables the evaluation of plant health and vegetation areas, contributing to a more comprehensive assessment of the effectiveness and impacts of NbS interventions.

## 2. Materials and Methods

### 2.1. Territorial Context of the Monitored Areas

In the context of the UPPER project, the monitored areas—sited in the Latina municipality (Latina, Italy)—before and during the intervention, as shown in Figure 3, are as follows: (a) CAMPO BOARIO (area designed as Productive Park), (b) MERCATO (area designed as Productive Park), (c) AUTOLINEE (area designed as Demonstration Site), (d) PIAZZA ILARIA ALPI (area designed as Demonstration Site), (e) VIA GOYA (area designed as Demonstration Site) and (f) VIA LEGNANO (area designed as Demonstration Site).

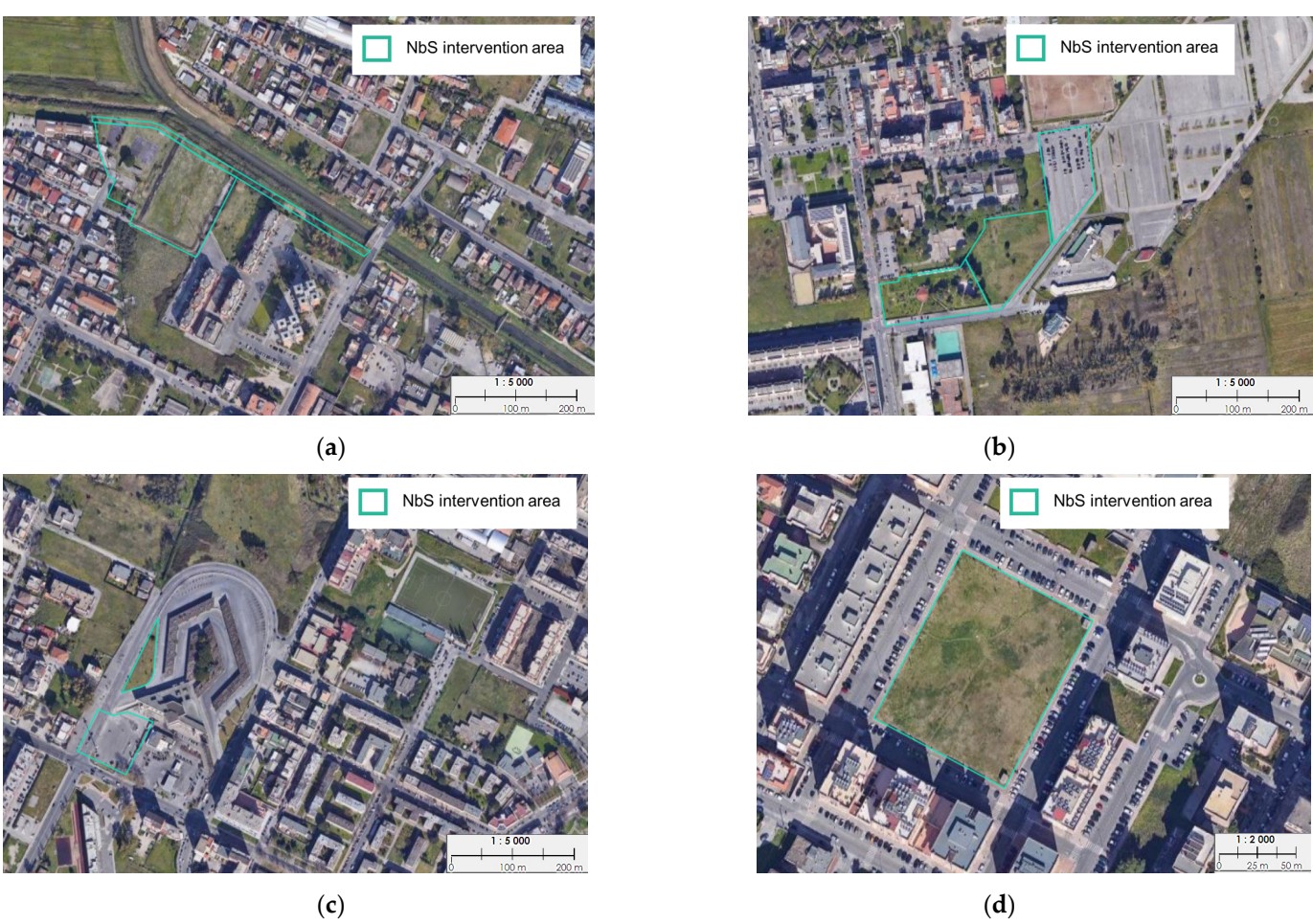

(**a**)

(**b**)

(**c**)

(**d**)

**Figure 3.** *Cont.*

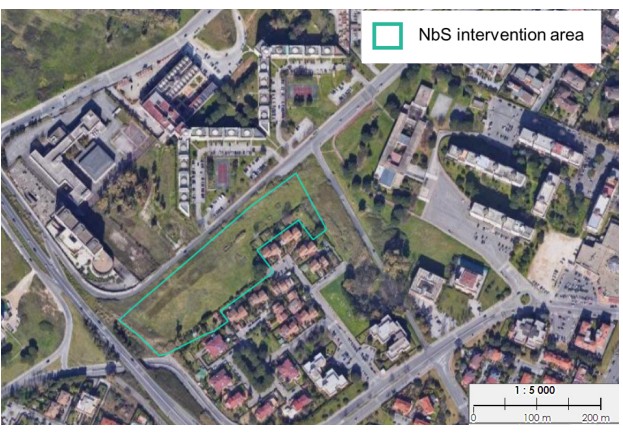

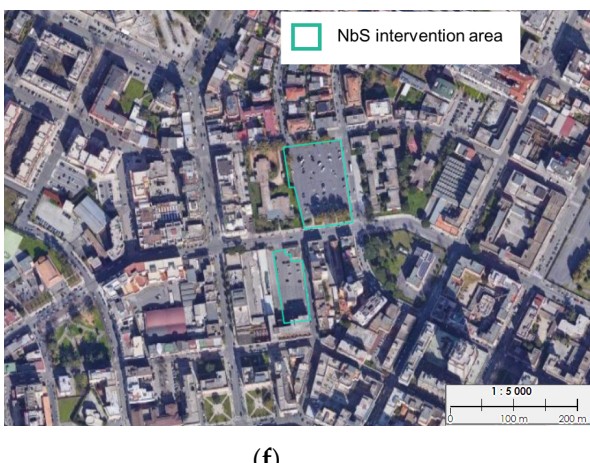

(**e**)                                                                              (**f**)

**Figure 3.** Territorial context of the planned NbS intervention areas within the UPPER project (2020): CAMPO BOARIO (**a**), MERCATO (**b**), AUTOLINEE (**c**), PIAZZA ILARIA ALPI (**d**), VIA GOYA (**e**) and VIA LEGNANO (**f**). Maps were extracted and elaborated within the WebGIS Holding Interactive Platform (WHiP) (https://www.cersites.gter.it/webgis_lpdt/lizmap/www/index.php/view/map/?repository=lab&project=UPPER_WHIP (accessed on 17 November 2022)).

The schematic diagram of UPPER project interventions and a summary description of the project elements are available in the Supplementary Materials for the following sites: (i) VIA GOYA, (ii) PIAZZA ILARIA ALPI, (iii), (iv) AUTOLINEE (VIA ROMAGNOLI), (v) VIA LEGNANO (VIA NEGHELLI and VIA LEPANTO) and (vi) MERCATO (P.LE DEI MERCANTI).

*2.2. Proximal Sensing Monitoring*

To obtain fundamental information regarding the most representative plant communities in the area, which serves as a reference for evaluating the effects of intervention success, non-invasive and non-destructive analyses were carried out. These analyses involved spectroscopic investigations that focused on assessing the chemical and physical characteristics of the leaf surfaces of selected tree individuals, which were considered as reference points. The investigations employed innovative monitoring techniques based on digital spectrophotometry in the NIR range (i.e., 1000–1700 nm). This range of wavelengths allows for the measurement of various leaf properties related to photosynthetic pigments, water content, and structural features [32]. By utilizing these advanced spectroscopic methods, valuable insights into the physiological status and health of the plant individuals were obtained without causing any harm or disruption to their overall well-being.

These non-invasive and non-destructive analyses provide a robust foundation for understanding the baseline characteristics of the plant communities in the area. They serve as valuable reference points for evaluating the effectiveness of interventions and assessing any changes or improvements brought about by the interventions.

2.2.1. The Proximal Sensing Device: Portable NIR Spectrometer

For the acquisition of reflectance spectra in the NIR range, the portable spectrometer MicroNIR 1700 was used (JDSU Corporation, Milpitas, CA, USA). This portable spectrometer is capable of acquiring reflectance spectra in the electromagnetic spectrum range from 950 to 1650 nm. The MicroNIR is based on the use of a linear variable filter (LVF) as a dispersive element, coupled with a 128-pixel InGaAs photodiode array. This portable instrument can be connected to a PC via a USB 2.0 cable and incorporates two incandescent lamps (tungsten filament).

The calibration procedure for the JDSU MicroNIR™ portable spectrometer followed the procedure described as follows.

The calibration white reference ($W_i$) was acquired using a Spectralon™ standard (with a nominal reflectance of 99%). The dark current ($D_i$) was acquired by covering the sensor spot with a black plastic (i.e., polyethylene) cap. The raw spectrum ($R_{0i}$) was then calculated as the reflectance spectrum ($R_i$) using the equation:

$$R_i = \frac{R_{0i} - D_i}{W_i - D_i} \tag{1}$$

The instrument calibration procedure and spectrum acquisition can be performed using MicroNIR Pro software (Ver. 1.2; JDSU Corporation, Milpitas, CA, USA). The acquired spectra in csv format are then exported into MATLAB (MathWorks, Inc., Natick, MA, USA).

### 2.2.2. Analyzed Samples, Spectra Collection and Handling

The acquisition of spectral information was carried out in situ on native plants already present in the areas planned for NbS interventions. In greater detail, the plants studied by area (as shown in Figure 4) are listed in Table 1.

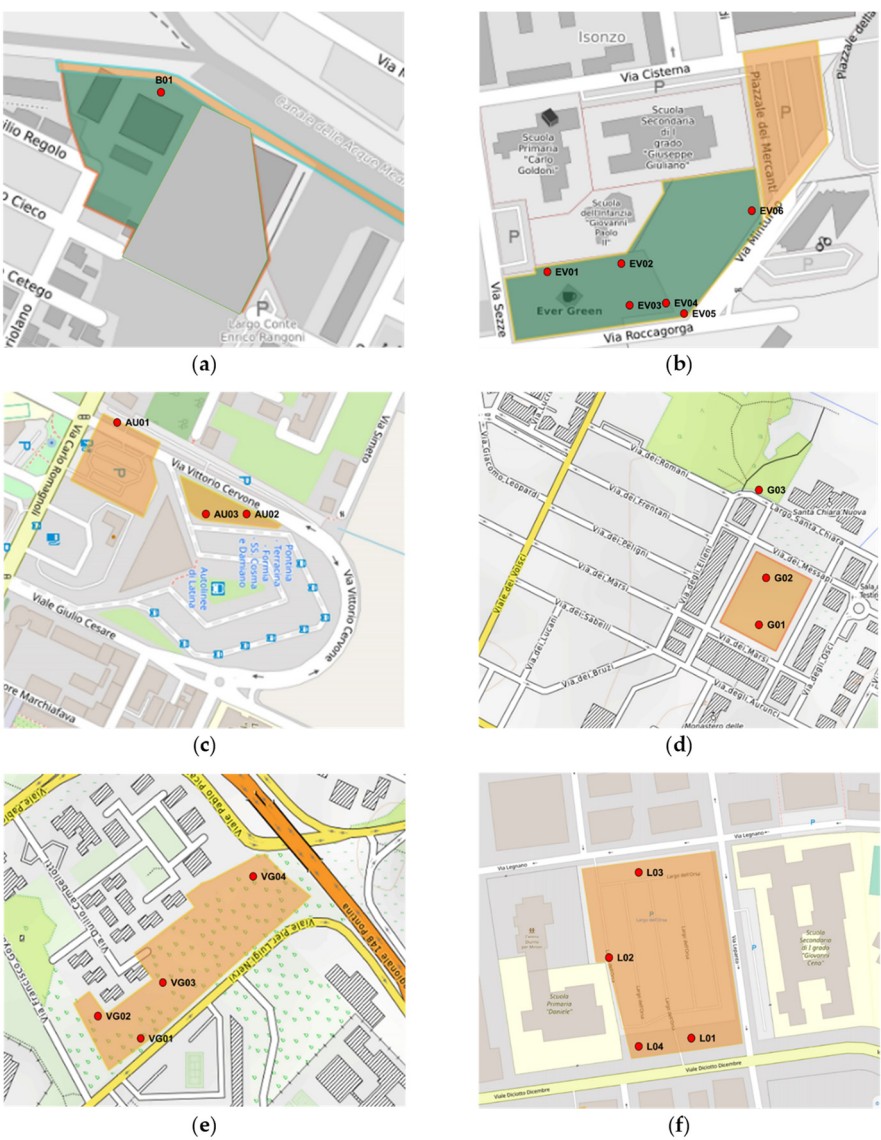

**Figure 4.** Schematic localization of the acquired plants (red dots) prior to NbS planned intervention (2020) of the sites: CAMPO BOARIO (**a**), MERCATO (**b**), AUTOLINEE (**c**), PIAZZA ILARIA ALPI (**d**), VIA GOYA (**e**), and VIA LEGNANO (**f**).

**Table 1.** Analyzed plants in UPPER sites.

| UPPER Sites | Analyzed Plants |
| --- | --- |
| CAMPO BOARIO | B01—*Junglans* L. (walnut). |
| MERCATO | EV01—*Laurus nobilis* L. (laurel), EV02—*Ulmus* L. (elm), EV03—*Quercus ilex* L. (evergreen oak), EV04—*Laurus nobilis* L. (laurel), EV05—*Celtis australis* L. (honeyberry), EV06—*Celtis australis* L. (honeyberry). |
| AUTOLINEE | AU01—*Eucalyptus obliqua* L'Hér. (Tasmanian oak), AU02—*Nerium oleander* L. (oleander), AU03—*Prunus spinosa* L. (blackthorn). |
| PIAZZA ILARIA ALPI | G01—*Quercus Cerris* L. (Turkey oak), G02—*Quercus ilex* L. (evergreen oak), G03—*Eucalyptus obliqua* L'Hér. (Tasmanian oak). |
| VIA GOYA | VG01—*Arundo donax* L. (Arundo), VG02—*Robinia pseudoacacia* L. (black locust), VG03—*Populus alba* L. (white poplar), VG04—*Arundo donax* L. (giant cane). |
| VIA LEGNANO | L01—*Eucalyptus obliqua* L'Hér. (Tasmanian oak), L02—Laurus nobilis (laurel), L03—*Eucalyptus obliqua* L'Hér. (Tasmanian oak), L04—*Ulmus minor* Mill. (field elm). |

Approximately 5 reflectance spectra were acquired for 4 different leaves on the same plant. From the averaged spectra for each sample, proximal sensing indices were subsequently calculated.

The method used for interpolating certain reflectance values, necessary for calculating the indices, was performed through weighted averaging. The linear interpolation between consecutive data points, denoted as $(R_a, R_b)$ and $(\lambda_a, \lambda_b)$, is given by:

$$R_c = \frac{\lambda_c(R_b - R_a) + \lambda_b R_a - \lambda_a R_b}{\lambda_b - \lambda_a} \qquad (2)$$

In Equation (2), $R_c$ represents the interpolated reflectance point at wavelength $\lambda_c$, within the range $[R_a, R_b]$. The interpolation process calculates the reflectance value $R_c$ by considering the relative positions of $R_a$ and $R_b$ and their respective wavelengths, $\lambda_a$ and $\lambda_b$. The weights, $\lambda_a$ and $\lambda_b$, represent the proportions assigned to each data point in the interpolation.

By employing this weighted averaging technique, the method estimates the reflectance values at specific wavelengths that may not be directly measured but are required for calculation of proximal sensing indices.

2.2.3. Proximal Sensing Indices Computing

Proximal sensing indices were evaluated from reflectance spectra. These indices are based on the corresponding remote sensing indices, providing information about vegetation status and estimates related to water content and Leaf Area Index (LAI).

In more detail, the indices that were calculated from the collected reflectance spectra in the NIR range with the MicroNIR are:

The Simple Ratio 1058/1148, also defined as RVI$_{hyp}$, a variant of the ratio vegetation index, is a vegetation index, formulated by [33], computed by dividing the reflectance at wavelength 1058 nm by the reflectance at 1148 nm (Equation (3)). The RVI$_{hyp}$ is related to the water content at the canopy level. A higher RVI$_{hyp}$ value implies healthier, more vigorous plants, whereas a lower value may indicate stress or less healthy vegetation.

The Simple Ratio 1193/1126, also defined as water content (WC), is a reflectance bands ratio defined by [34]. This band ratio (Equation (4)) is specifically related to water content in plants. A higher WC value indicates a higher water content in the plant, whereas a lower value suggests a lower water content.

The Normalized Difference 1094/1205, also defined as the Leaf Water Vegetation Index 2 (LWVI$_2$), is a Normalized Difference Water Index (NDWI) variant and was formulated by [35]. This reflectance band difference ratio is related to leaf water content and is calculated as reported by Equation (5).

The Normalized Difference Water Index–Hyperion (NDWI$_{\text{Hyp}}$), or Normalized Difference 1070/1200, is a reflectance band difference ratio formulated by [36] as reported in Equation (6). The NDWI$_{\text{Hyp}}$ was formulated to estimate canopy water content since the reflectance at 1070 nm is strongly influenced by the chlorophyll content of the leaves, while the reflectance at 1200 nm is more affected by the water content of the leaves.

Moreover, four reflectance bands were considered: Single Band 970 ($R_{970nm}$), Single Band 1200 ($R_{1200nm}$), Single Band 1400 ($R_{1400nm}$), and Single Band 1450 ($R_{1450nm}$). In more detail, $R_{970nm}$ is a reflectance band related to vegetation water and starch [37] and $R_{1400nm}$ can be attributed to the water of the vegetation. The $R_{1200nm}$ and $R_{1450nm}$ are reflectance bands related to vegetation water, cellulose, lignin, and starch.

$$\text{RVI}_{\text{hyp}} = \frac{R_{1058\text{nm}}}{R_{1148\text{nm}}} \tag{3}$$

$$WC = \frac{R_{1193nm}}{R_{1126nm}} \tag{4}$$

$$LWVI_2 = \frac{R_{1094nm} - R_{1205nm}}{R_{1094nm} + R_{1205nm}} \tag{5}$$

$$NDWI_{Hyp} = \frac{R_{1070nm} - R_{1200nm}}{R_{1070nm} + R_{1200nm}} \tag{6}$$

*2.3. Remote Sensing Monitoring*

To assess the change from the ex ante state (or baseline state) to the intervention state of the UPPER areas in terms of plant health and vegetation evaluation, a study using satellite imagery was conducted. The areas of interest (AOIs) were defined based on the intervention areas to evaluate the mapping of vegetation indices such as the Normalized Difference Vegetation Index (NDVI), Normalized Difference Water Index (NDWI), and Normalized Difference Built-up Index (NDBI). Using the historical records of Sentinel-2 L2A satellite imagery, the average calculation of the NDVI and NDWI indices was performed within the AOIs. This calculation was carried out using all available Sentinel-2 L2A satellite maps from the last few years (2015–2023).

The average value and standard deviation for each NDVI (and NDWI) observation were calculated considering data from all years within a specific timeframe. Specifically, the average value and standard deviation were calculated for each year, and then, these values were separately averaged, over the entire period (2015–2022), for each site.

The temporal series of these indices enable a comparison between the ex-ante state of NbS interventions and the ongoing interventions within the areas. By analyzing the trends and changes in the NDVI and NDWI indices over time, an evaluation of the impact of the NbS on plant health and vegetation conditions can be made.

2.3.1. The Sentinel-2 L2A Satellite

Sentinel-2 is a high-resolution and wide-swath multispectral imaging mission that supports Copernicus Land Monitoring applications such as vegetation, soil, and water cover monitoring, as well as inland waterways and coastal observation [38]. The geoinformation obtained from Sentinel-2 data can be analyzed and utilized for various applications, including land use planning, agro-environmental monitoring, water monitoring, forest and vegetation monitoring, natural resource monitoring, and crop monitoring. Regarding the temporal resolution, the individual Sentinel-2 satellites have a revisit frequency of 10 days, while the combined revisit frequency of the constellation is 5 days.

The multispectral instrument (MSI) of Sentinel-2 samples 13 spectral bands at different spatial resolutions. Spatial resolution of the MSI instrument refers to the size of the ground elementary area from which electromagnetic energy is detected (pixel). In more detail, Sentinel-2 has different spatial resolutions depending on the spectral bands:

(i)     10 m spatial resolution (B2 (490 nm), B3 (560 nm), B4 (665 nm) and B8 (842 nm));
(ii)    20 m spatial resolution (B5 (705 nm), B6 (740 nm), B7 (783 nm), B8a (865 nm), B11 [1610 nm] and B12 [2190 nm]);
(iii)   60 m spatial resolution (B1 (443 nm), B9 (940 nm) and B10 (1375 nm)).

### 2.3.2. Satellite Data Analysis and Remote Sensing-Derived Indices

The software application used for processing satellite data is Landviewer (https://eos.com/landviewer/; EOS Data Analytics, Inc., Mountain View, CA, USA). The satellite images (Figure 5) utilized for identifying the Areas of Interest (AOIs) and calculating vegetation index maps are from 20 May 2022 (prior to NbS interventions) and 5 May 2023 (during the NbS interventions).

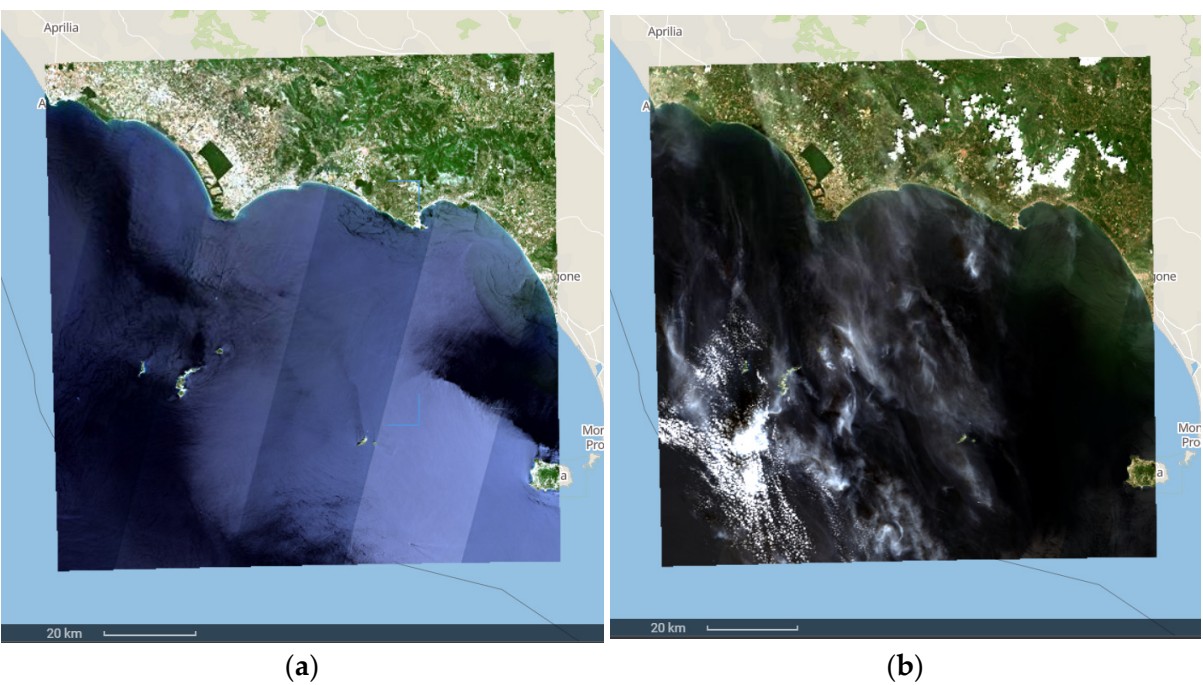

(**a**)                                                          (**b**)

**Figure 5.** Sentinel-2 L2A satellite images used for assessing the Areas of Interest (AOIs): (**a**) 20 May 2022 (prior NbS interventions) and (**b**) 5 May 2023 (during NbS interventions). Satellite images were retrieved using Landviewer (EOS Data Analytics, Inc., Mountain View, CA, USA).

In detail, the natural color satellite maps are derived from the B4 (665 nm), B3 (560 nm), and B2 (490 nm) bands. The selected AOIs of city centers and NbS sites are shown in Figures 6 and 7.

Starting from the collected satellite images, the remote sensing indices NDVI, NDWI, and NDBI were evaluated.

**NDVI.** Healthy plants exhibit high reflectance in the NIR range between 0.7 and 1.3 µm and absorb significantly in the red range of Vis. It is for this reason that the red and NIR bands are used to calculate the NDVI.

The NDVI was evaluated using the B8a (865 nm) and B4 (665 nm) bands of Sentinel-2:

$$NDVI = \frac{(NIR - Red)}{(NIR + Red)} = \frac{(R_{865\ nm} - R_{665\ nm})}{(R_{865\ nm} + R_{665\ nm})} \quad (7)$$

The value of the NDVI index comes from its capacity to reveal details about the quantity of chlorophyll and photosynthetic activity of plants. The ratio between these two spectral bands can reveal the presence and amount of plant biomass since chlorophyll heavily absorbs red light and reflects NIR light.

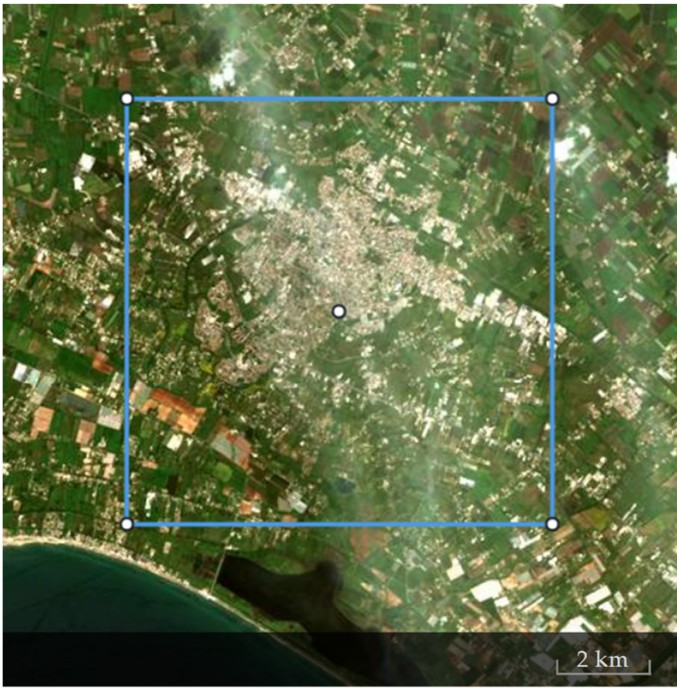

**Figure 6.** AOI of the Latina city center.

**NDWI.** Bodies of water generally have high reflectance in the blue spectral range (0.4–0.5 μm) compared to green (0.5–0.6 μm) and red (0.6–0.7 μm). Based on these principles, The NDWI defined by [39] allows for the detection and monitoring of water bodies by measuring the difference in reflectance between the green and near-infrared bands.

The NDWI was calculated using the B3 (560 nm) and B8 (842 nm) bands as follows:

$$NDWI = \frac{(GREEN - NIR)}{(GREEN + NIR)} = \frac{(R_{560\ nm} - R_{842\ nm})}{(R_{560\ nm} + R_{842\ nm})} \tag{8}$$

This index is a valuable tool for assessing water presence and monitoring changes in aquatic environments. Generally, the NDWI assumes values greater than 0.3 when in the presence of water.

**NDBI.** NDBI is widely used in remote sensing and urban studies to identify and assess urban areas and their spatial patterns [40]. This index allows us to highlight urban areas with higher reflectance in the shortwave infrared (SWIR) compared to the NIR region. The NDBI was evaluated using the B11 (1610 nm) and B8 (842 nm) bands:

$$NDBI = \frac{(SWIR - NIR)}{(SWIR + NIR)} = \frac{(R_{1610\ nm} - R_{842\ nm})}{(R_{1610\ nm} + R_{842\ nm})} \tag{9}$$

The NDBI values range from −1 to +1. Negative values of NDBI represent water bodies, while higher positive values indicate built-up areas.

Subsequently, based on the historical Sentinel-2 L2A satellite maps, the average calculation of the NDVI and NDWI indices was performed within the Areas of Interest (AOIs). The calculation of these indices was carried out using all available Sentinel-2 L2A satellite maps from the past few years (2015–2023).

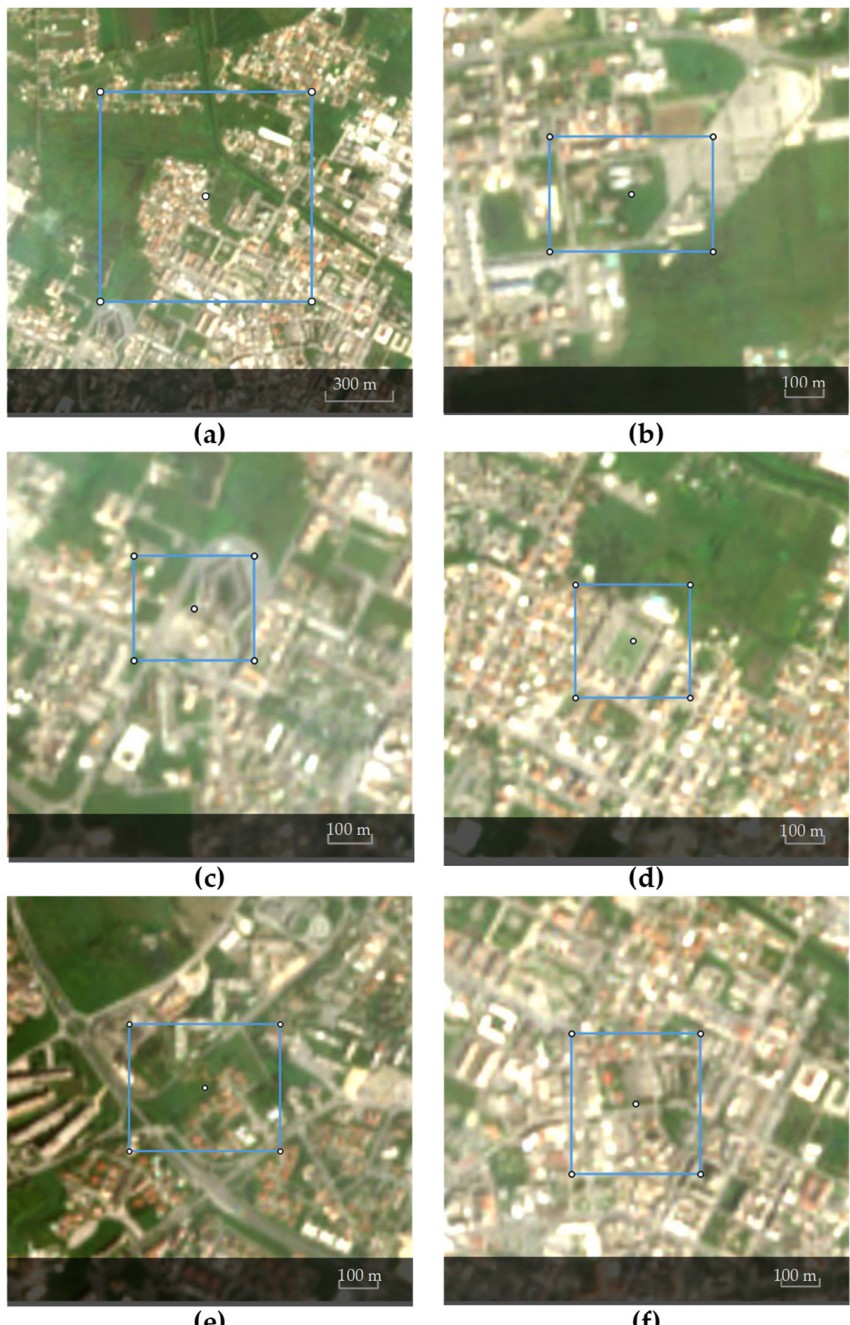

**Figure 7.** AOIs of planned NbS intervention areas within the UPPER project (5 May 2023): CAMPO BOARIO (**a**), MERCATO (**b**), AUTOLINEE (**c**), PIAZZA ILARIA ALPI (**d**), VIA GOYA (**e**), and VIA LEGNANO (**f**).

### 2.4. Meteo-Climatic Monitoring

Since each year is unique in terms of the phenological characteristics, which are influenced by various factors such as climate fluctuations and meteorological conditions spanning both long and short timeframes, meteorological data were taken into account when evaluating and comparing the different indices [41]. In more detail, the 2015–2023 time series of mean daily precipitation (mm), maximum daily temperature (°C), average daily temperature (°C), minimum daily temperature (°C), and daily relative humidity (%) were considered. These variables were extracted from a weather-climate station located in Borgo Carso (Cisterna di Latina, Latina). Data are accessible at the Agrometeorological Integrated Service of Lazio Region (SIARL—*Servizio Integrato Agrometeorologico della Regione*

*Lazio*; https://www.siarl-lazio.it/ (accessed on 30 May 2023)) website. In order to proceed with the data analysis, daily precipitation, daily temperature, and daily relative humidity were averaged based on the number of observations available for each year.

Moreover, two different drought monitoring products were adopted to evaluate the rainfall deficits on the vegetation of Latina City [42]. Monthly data of spatial average Soil Moisture Index (SMI) and Standardized Precipitation Index for an accumulation period of 3 months (SPI-3) for NUTS (nomenclature of territorial units for statistics) 3 level region IT44 Latina were retrieved from the Copernicus European Drought Observatory (EDO) website (available at: https://edo.jrc.ec.europa.eu/edov2/php/index.php?id=1000 (accessed on 30 May 2023)) of the EC Joint Research Centre. Monthly data for the time period January 2015–May 2023 were averaged for each year. The SMI, derived from 6-hourly LISFLOOD modelled soil moisture in the top two soil layers [43], is an index that can be used to assess the moisture content in the soil [44]. It provides information about how much water is present in the top layer of the soil, which is crucial for vegetation growth and monitoring drought conditions. Higher SMI values indicate wetter soil conditions, implying that there is enough moisture in the soil to support plant growth. On the other hand, a lower SMI score denotes drier soil conditions, which may indicate that the vegetation is experiencing drought stress. The SPI-3 is a widely adopted metric for identifying and characterizing meteorological droughts, which refer to prolonged periods of below-average rainfall in a specific area [45]. It assesses deviations in precipitation levels by comparing observed total precipitation over a selected accumulation period (i.e., three months) with the historical rainfall data for that same period. SPI-3 values falling below $-1.0$ indicate a deficit in rainfall (drier conditions), whereas SPI-3 values above 1.0 indicate an excess of rainfall (wetter conditions). The magnitude of the drought is inversely proportional to the SPI value.

### 2.5. Exploratory, Statistical Data, and Multivariate Regression Analyses

Principal Component Analysis (PCA) was adopted to explore meteorological data and check whether observed differences between 2022 and 2023 are not solely due to long-term variability. PCA is a statistical technique, usually adopted to reduce the number of dimensions in complex data and to observe trends and patterns of the analyzed data [46]. PCA converts original variables into Principal Components (PCs), which are orthogonal to each other, meaning that they are uncorrelated. PCs are linear combinations of the original variables. Autoscale was adopted as a pre-processing algorithm for the data undergoing PCA.

To assess potential differences among certain examined variables, appropriate statistical analyses such as correlation analysis, *t*-tests, or Analysis of Variance (ANOVA) were conducted [47,48]. The threshold for statistical significance was set at 0.05.

The Partial Least Squares (PLS) regression method was tested to predict vegetation indices. PLS is a technique for relating two data matrices, X (input) and Y (output), by a linear multivariate model [49,50]. Such a technique was adopted in order to evaluate the specific impact of the intervention while considering also the contributions of climate variations.

In the first step of this analysis, PLS regression models were trained for each investigated UPPER site. In each site model, data from the years 2015 to 2022 of the considered site was used for training. This included climate variables as input X and vegetation indices (NDVI and NDWI) as output Y. Venetian Blinds (VBs) was used as cross-validation algorithm in order to choose the right complexity of each model by assessing the right number of Latent Variables (LVs). Then, an inference for the year 2023 was performed for each site. Data for the year 2023 was used as the test set and only the climate variables were considered as inputs. The trained model was adopted to predict the vegetation indices of each site. The goodness of fit of the data in the regression models was assessed with the coefficient of determination, $R^2$. The Root Mean Square Error (RMSE) was used as a measure for the average deviation of the predicted values from the actual values. The bias was used as a metric for the systematic error/offset in the predictions compared to the actual values.

These parameters were evaluated for the calibration (C) and the cross-validation phases of the modelling. Finally, in the validation phase, the actual observations for 2023 were compared against the model-predicted values. Statistical tests (*t*-tests) were performed on the NDVI and NDWI actual values and predictions to determine if there are significant differences between the predicted and actual values.

All the analyses were performed using PLS Toolbox (Ver 9.0; Eigenvector Research, Inc., Manson, WA, USA) and Statistics and Machine Learning Toolbox (Ver. 12.3; MathWorks, Inc., Natick, MA, USA), and BANSHEE (Ver. 10; [51]) running in MATLAB and an Excel (Ver. 2308; Microsoft Corporation, Redmond, WA, USA) spreadsheet.

## 3. Results and Discussion

### 3.1. In-Field Proximal Sensing Indices Ex Ante NbS Interventions

The average reflectance spectra of each analyzed plant are shown in Figure 8. The figure shows characteristics and absorption patterns that are associated with various biochemical and structural components. In more detail, a significant drop in reflectance can be observed between 1400–1500 nm. This is caused by the strong absorption of water molecules in the leaf tissues, indicating the presence of water content in the plant [37]. Beyond 1500 nm, the reflectance gradually increases due to the presence of different chemical compounds, such as cellulose, lignin, and other structural components in the plant cell walls.

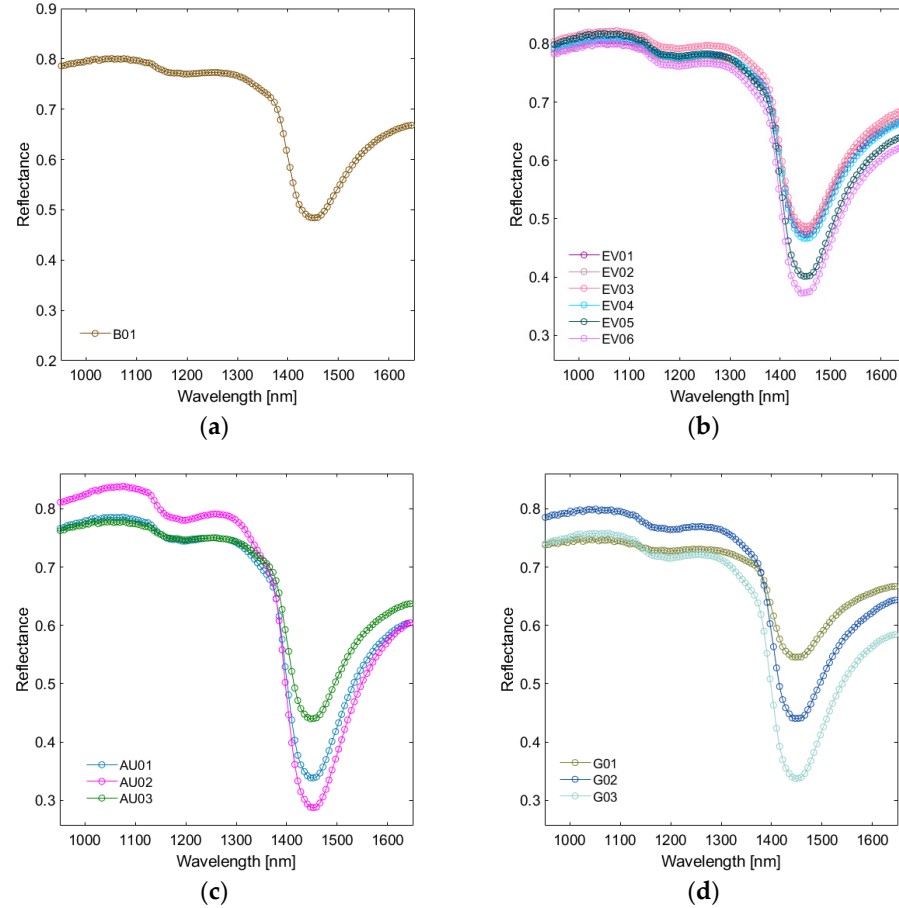

**Figure 8.** *Cont*.

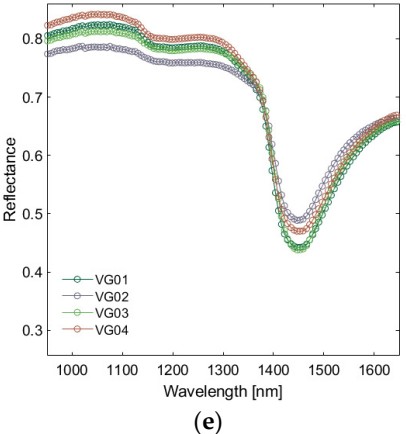

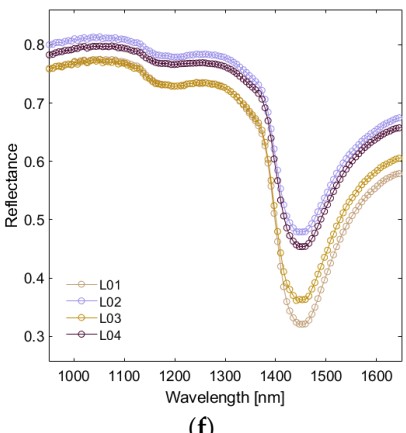

**Figure 8.** Mean reflectance spectra of the acquired plants prior to NbS planned intervention (2020): CAMPO BOARIO (**a**), MERCATO (**b**), AUTOLINEE (**c**), PIAZZA ILARIA ALPI (**d**), VIA GOYA (**e**), and VIA LEGNANO (**f**).

The derived indices as resulting from the analysis of each plant are reported in Table 2, while in Table 3 are reported the indices averaged according to each site.

**Table 2.** Indices' average values evaluated starting from the mean reflectance spectra of each plant (ex ante NbS interventions).

| Sample ID | Plants | LAIDI | $RVI_{hyp}$ | WC | $LWVI_2$ | $NDWI_{H}$ | $R_{970\ nm}$ | $R_{1200\ nm}$ | $R_{1250\ nm}$ | $R_{1400\ nm}$ | $R_{1450\ nm}$ |
|---|---|---|---|---|---|---|---|---|---|---|---|
| B01 | Walnut | 0.965 | 1.026 | 0.973 | 0.018 | 0.019 | 0.791 | 0.770 | 0.773 | 0.604 | 0.484 |
| EV01 | Laurel | 0.968 | 1.025 | 0.971 | 0.018 | 0.019 | 0.796 | 0.775 | 0.779 | 0.606 | 0.473 |
| EV02 | Elm | 0.969 | 1.024 | 0.974 | 0.016 | 0.018 | 0.791 | 0.772 | 0.775 | 0.605 | 0.481 |
| EV03 | Evergreen oak | 0.970 | 1.025 | 0.972 | 0.017 | 0.018 | 0.810 | 0.792 | 0.797 | 0.619 | 0.487 |
| EV04 | Laurel | 0.966 | 1.026 | 0.969 | 0.019 | 0.020 | 0.797 | 0.776 | 0.781 | 0.599 | 0.466 |
| EV05 | Honeyberry | 0.957 | 1.033 | 0.967 | 0.022 | 0.024 | 0.805 | 0.779 | 0.782 | 0.562 | 0.401 |
| EV06 | Honeyberry | 0.956 | 1.033 | 0.964 | 0.023 | 0.024 | 0.788 | 0.761 | 0.766 | 0.530 | 0.373 |
| AU01 | Tasmanian oak | 0.955 | 1.035 | 0.960 | 0.025 | 0.026 | 0.773 | 0.744 | 0.750 | 0.508 | 0.339 |
| AU02 | Oleander | 0.945 | 1.046 | 0.944 | 0.034 | 0.035 | 0.816 | 0.780 | 0.790 | 0.478 | 0.288 |
| AU03 | Blackthorn | 0.964 | 1.026 | 0.971 | 0.018 | 0.020 | 0.768 | 0.747 | 0.750 | 0.577 | 0.440 |
| G01 | Turkey oak | 0.976 | 1.017 | 0.983 | 0.012 | 0.013 | 0.741 | 0.727 | 0.730 | 0.629 | 0.546 |
| G02 | Evergreen oak | 0.962 | 1.026 | 0.969 | 0.020 | 0.021 | 0.790 | 0.764 | 0.769 | 0.586 | 0.441 |
| G03 | Tasmanian oak | 0.952 | 1.038 | 0.958 | 0.026 | 0.028 | 0.746 | 0.715 | 0.721 | 0.489 | 0.338 |
| VG01 | Arundo | 0.956 | 1.035 | 0.963 | 0.023 | 0.024 | 0.810 | 0.784 | 0.787 | 0.558 | 0.442 |
| VG02 | Black locust | 0.966 | 1.025 | 0.976 | 0.016 | 0.017 | 0.779 | 0.758 | 0.759 | 0.605 | 0.489 |
| VG03 | White poplar | 0.963 | 1.028 | 0.970 | 0.019 | 0.021 | 0.802 | 0.779 | 0.783 | 0.587 | 0.438 |
| VG04 | Arundo | 0.953 | 1.040 | 0.965 | 0.024 | 0.026 | 0.828 | 0.799 | 0.802 | 0.579 | 0.470 |
| L01 | Tasmanian oak | 0.948 | 1.039 | 0.955 | 0.028 | 0.030 | 0.764 | 0.729 | 0.735 | 0.490 | 0.321 |
| L02 | Laurel | 0.963 | 1.026 | 0.970 | 0.020 | 0.020 | 0.804 | 0.779 | 0.783 | 0.606 | 0.479 |
| L03 | Tasmanian oak | 0.948 | 1.037 | 0.961 | 0.026 | 0.028 | 0.763 | 0.729 | 0.734 | 0.505 | 0.363 |
| L04 | Field elm | 0.964 | 1.027 | 0.972 | 0.018 | 0.020 | 0.788 | 0.766 | 0.769 | 0.588 | 0.454 |

Min          Max

**Table 3.** Averaged in-field indices for UPPER sites (ex ante NbS interventions).

| UPPER Site | LAIDI | $RVI_{hyp}$ | WC | $LWVI_2$ | $NDWI_{Hyp}$ |
|---|---|---|---|---|---|
| CAMPO BOARIO | 0.965 | 1.026 | 0.973 | 0.018 | 0.019 |
| MERCATO | 0.964 | 1.028 | 0.970 | 0.019 | 0.020 |
| AUTOLINEE | 0.955 | 1.036 | 0.958 | 0.026 | 0.027 |
| PIAZZA ILARIA ALPI | 0.963 | 1.027 | 0.970 | 0.020 | 0.021 |
| VIA GOYA | 0.960 | 1.032 | 0.968 | 0.020 | 0.022 |
| VIA LEGNANO | 0.956 | 1.032 | 0.964 | 0.023 | 0.024 |

Min          Max

As shown in Table 2, the LAIDI values range from 0.945 to 0.976, indicating variations in the leaf area density among the sites. Higher LAIDI values, suggesting a denser vegetation or dense leaf area, are obtained for samples G01 (Turkey oak; 0.97), EV02 (elm; 0.969), and EV03 (evergreen oak; 0.970). The $RVI_{hyp}$ values range from 1.017 to 1.046. A higher $RVI_{hyp}$ score suggests healthier vegetation. Samples with higher $RVI_{hyp}$ scores are AU02 (oleander; 1.046), L03 (Tasmanian oak; 1.037), and EV05 (honeyberry; 1.033). The WC values range from 0.944 to 0.983, indicating variations in the water content within the vegetation. Higher WC values suggest higher moisture levels, which can be beneficial for plant growth. In this case, the highest values of WC were obtained for G01 (Turkey oak; 0.983), followed by EV02 (elm; 0.974) and EV03 (evergreen oak; 0.972). Variations in the vegetation's ability to retain water are indicated by $LWVI_2$ values, which range from 0.012 to 0.034. Higher $LWVI_2$ values, as seen in samples AU02 (oleander; 0.035), L01 (Tasmanian oak; 0.030), and VG04 (Arundo; 0.026) suggest better water retention capacity. Finally, the $NDWI_{Hyp}$ values ranging from 0.013 to 0.035 indicate variations in water content within the vegetation. Higher $NDWI_{Hyp}$ values suggest higher water content, which can be an indicator of healthier vegetation. The highest $NDWI_{Hyp}$ values were obtained for AU02 (oleander; 0.035), L01 (Tasmanian oak; 0.030), and VG04 (Arundo; 0.026).

From Table 3, it appears that among the analyzed sites, CAMPO BOARIO stands out as having the best environmental performance. It exhibits a higher LAIDI value of 0.965, indicating a denser vegetation cover. Additionally, the $RVI_{hyp}$ value of 1.026 suggests healthy vegetation, and the WC value of 0.973 indicates a moderate water content. These indicators collectively suggest that CAMPO BOARIO has a robust and healthy vegetation ecosystem, which is essential for environmental balance and ecosystem services.

However, CAMPO BOARIO's indices were evaluated on the basis of only one plant (B01). With LAIDI, $RVI_{hyp}$, WC, $LWVI_2$, and $NDWI_{Hyp}$ values of 0.963, 1.027, 0.970, 0.020, and 0.021, respectively, PIAZZA ILARIA ALPI demonstrates a strong environmental performance, ranking second in the classification. VIA GOYA follows closely with LAIDI, $RVI_{hyp}$, WC, LWVI2, and $NDWI_{Hyp}$ values of 0.960, 1.032, 0.968, 0.020, and 0.022, respectively, indicating a relatively good environmental performance.

With LAIDI, $RVI_{hyp}$, WC, $LWVI_2$, and $NDWI_{Hyp}$ values of 0.964, 1.028, 0.970, 0.019, and 0.020, respectively, the MERCATO site also demonstrates a positive environmental performance. VIA LEGNANO follows with LAIDI, $RVI_{hyp}$, WC, $LWVI_2$, and $NDWI_{Hyp}$ values of 0.956, 1.032, 0.964, 0.023, and 0.024, respectively, suggesting a moderate environmental performance. Finally, AUTOLINEE shows relatively lower environmental performance compared to other analyzed sites. It has a lower LAIDI value of 0.955, indicating a slightly less dense vegetation cover. The $RVI_{hyp}$ value of 1.036 suggests relatively healthier vegetation, but the WC value of 0.958 indicates a lower water content. Although AUTOLINEE still demonstrates some positive environmental attributes, such as healthy vegetation, the lower LAIDI and WC values imply potential limitations in terms of vegetation density and water availability, which can affect the overall environmental performance.

Further in-field investigations about the vegetation status of the UPPER sites after NbS interventions are programmed to be held in late 2023.

### 3.2. Remote Sensing Index Variations

This paragraph presents and discusses the changes in the NDVI and NDWI between 2022 and 2023, based on the available Sentinel-2 maps.

The time series 2015–2023 of these indices for each analyzed site are provided in the Supplementary Materials. Figures 9 and 10 display graphical representations of the NDVI and NDWI variations on 20 May 2022 (before NbS interventions) and 5 May 2023 (during NbS interventions), respectively. On the other hand, graphical representations of the NDBI variation for the two time points are available in the Supplementary Materials.

3.2.1. Variation in NDVI Values (2022–2023)

The highest NDVI mean value in 2022 was achieved for VIA GOYA (0.416), indicating relatively healthier vegetation compared to other areas. On the other hand, AUTOLINEE has the lowest mean value of 0.231 for the 2022 NDVI index. In 2023, the highest mean value for the NDVI index is VIA GOYA, with a value of 0.443. This indicates relatively healthier vegetation in VIA GOYA compared to the other areas. On the other hand, VIA LEGNANO has the lowest mean value (0.201) for the NDVI index in 2023, indicating poorer vegetation health compared to the other areas. The city center of Latina experienced a positive increase of 0.076 in the NDVI index over the course of one year (Table 4). Furthermore, all analyzed areas showed a positive increase in the NDVI index, except for VIA LEGNANO, where a slight decrease in the NDVI index was recorded. In this context, an increase in the NDVI index could indicate an increase in vegetation. This may be due to greater vegetation biomass, increased leaf production, or improved photosynthetic efficiency of plants. Generally, an increase in the NDVI index signifies a healthy ecosystem and high vegetation productivity. However, an excessively high increase in the NDVI index could be indicative of excessive vegetation growth, which could be a result of inadequate land management.

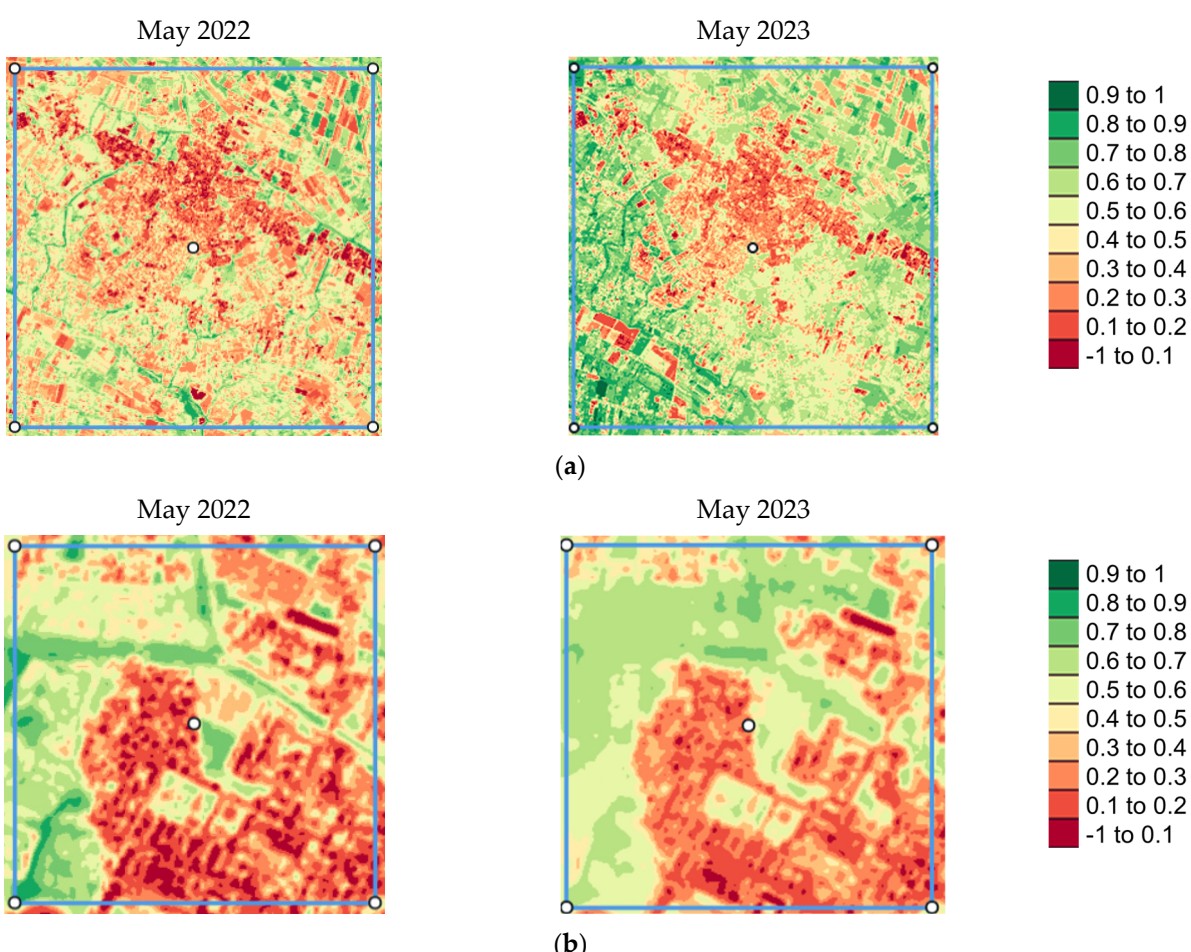

**Figure 9.** *Cont.*

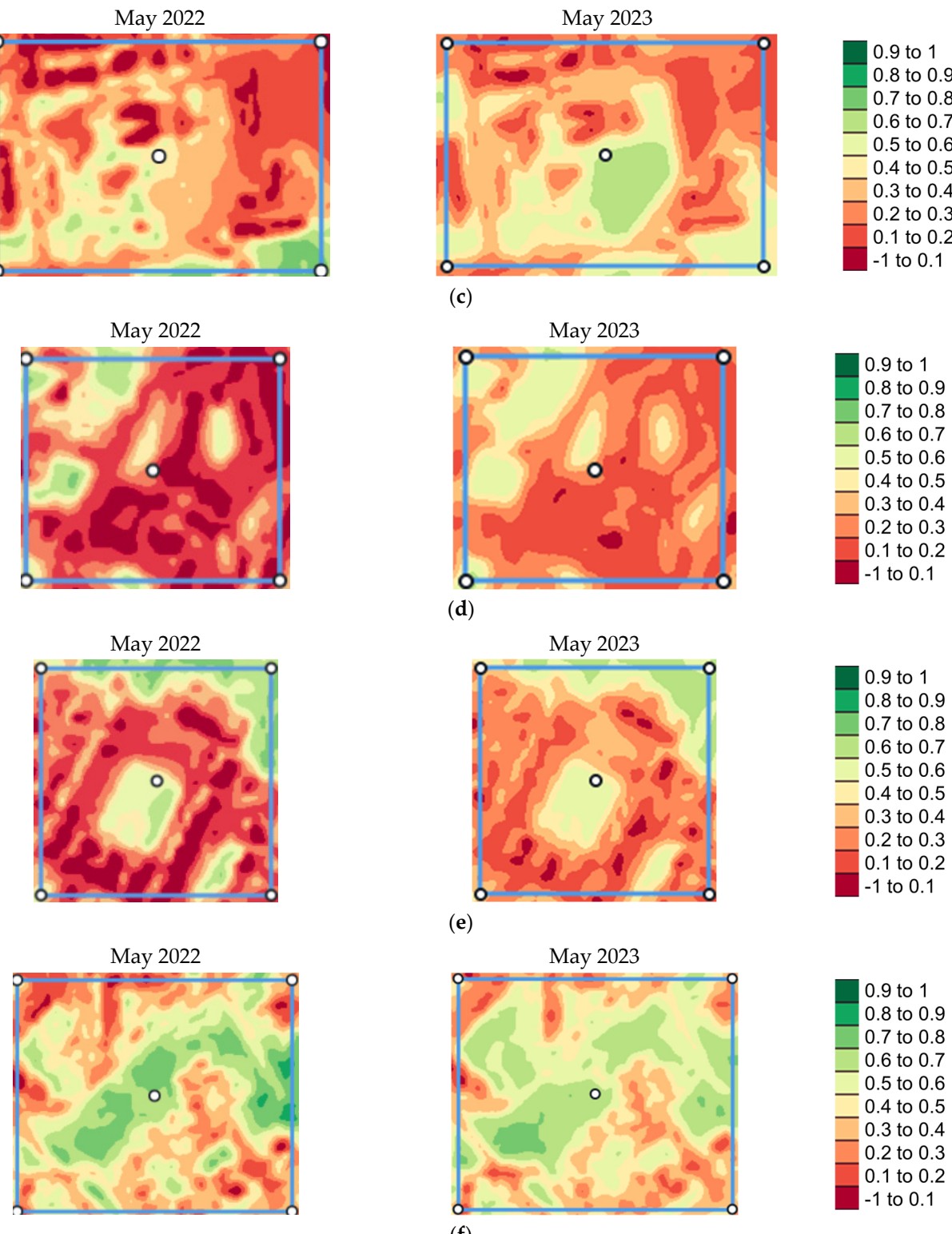

**Figure 9.** *Cont.*

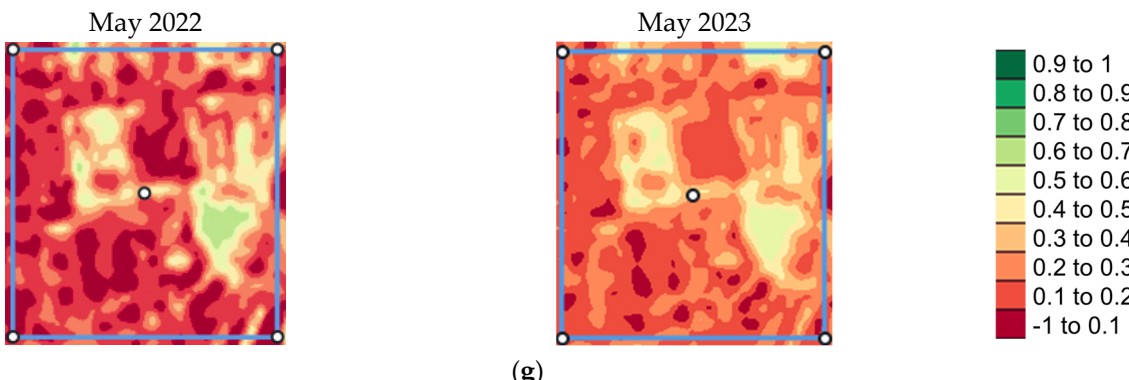

(**g**)

**Figure 9.** NDVI false color maps of 20 May 2022 (prior to NbS interventions) and 5 May 2023 (during NbS interventions): city center (**a**), CAMPO BOARIO (**b**), MERCATO (**c**), AUTOLINEE (**d**), PIAZZA ILARIA ALPI (**e**), VIA GOYA (**f**), and VIA LEGNANO (**g**).

**Figure 10.** *Cont.*

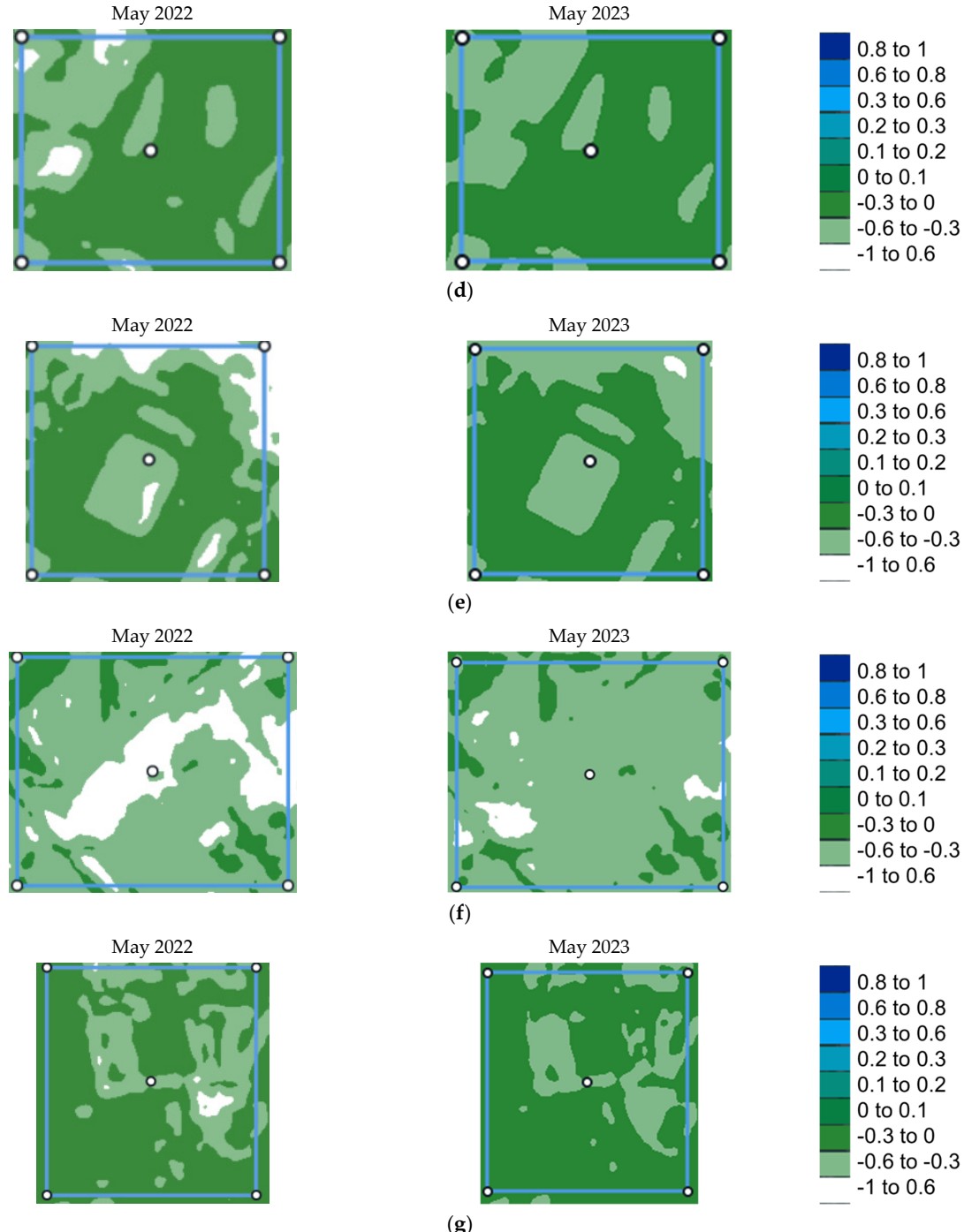

**Figure 10.** NDWI false color maps of 20 May 2022 (prior to NbS interventions) and 5 May 2023: city center (**a**), CAMPO BOARIO (**b**), MERCATO (**c**), AUTOLINEE (**d**), PIAZZA ILARIA ALPI (**e**), VIA GOYA (**f**), and VIA LEGNANO (**g**).

3.2.2. Variation in NDWI Values (2022–2023)

Regarding the NDWI index, a decrease is observed for the city center during the period of 2022–2023 (Table 4). A decrease in this index is also observed for the following areas: CAMPO BOARIO, MERCATO, AUTOLINEE, and VIA GOYA. On the other hand, an increase in the NDWI index is observed for VIA LEGNANO and PIAZZA ILARIA ALPI.

**Table 4.** NDVI and NDWI variations (2022–2023) for the city center of Latina and UPPER project sites.

| | NDVI | | | NDWI | | |
|---|---|---|---|---|---|---|
| | **2022** | **2023** | **Increment** | **2022** | **2023** | **Increment** |
| City center | 0.396 ± 0.164 | 0.472 ± 0.177 | +0.076 | −0.311 ± 0.130 | −0.362 ± 0.150 | −0.051 |
| CAMPO BOARIO | 0.382 ± 0.180 | 0.411 ± 0.198 | +0.028 | −0.280 ± 0.141 | −0.299 ± 0.125 | −0.019 |
| MERCATO | 0.326 ± 0.161 | 0.339 ± 0.183 | +0.013 | −0.252 ± 0.124 | −0.256 ± 0.144 | −0.004 |
| AUTOLINEE | 0.231 ± 0.161 | 0.253 ± 0.196 | +0.022 | −0157 ± 0.144 | −0.166 ± 0.166 | −0.009 |
| PIAZZA ILARIA ALPI | 0.300 ± 0.179 | 0.322 ± 0.211 | +0.021 | −0.208 ± 0.155 | −0.200 ± 0.193 | +0.009 |
| VIA GOYA | 0.416 ± 0.142 | 0.443 ± 0.167 | +0.025 | −0.326 ± 0.111 | −0.336 ± 0.140 | −0.010 |
| VIA LEGNANO | 0.206 ± 0.120 | 0.201 ± 0.159 | −0.006 | −0.154 ± 0.108 | −0.165 ± 0.096 | −0.011 |

An increase in the NDWI index indicates a higher water uptake by plants, which can be attributed to increased vegetation biomass, greater water availability, or improved hydraulic efficiency of plants. This increase may also be indicative of enhanced vegetation productivity, increased herbaceous biomass, and other indicators of ecosystem health. Additionally, an increase in the NDWI index can be influenced by weather conditions, soil quality, and water availability.

3.2.3. Considerations about NDVI and NDWI Variations (2022–2023)

The annual variations that occurred between 2022 and 2023 of the NDVI and NDWI for each analyzed site are reported in Table 4.

The NDVI in the city center indicates an increase of +0.076, thus showing an overall improvement in the health or density of the vegetation. However, a decrease in the water content of the plants is indicated by the NDWI's negative increment of −0.051.

In the CAMPO BOARIO site, the NDVI showed a positive increment of +0.028, indicating a slight improvement in vegetation health, while NDWI showed a shift of −0.019. While the NDWI in MERCATO shows a minimal shift of −0.004, the NDVI reveals a slightly positive increment of +0.013, indicating a rather steady vegetative condition but a slight reduction in water content. The NDVI in AUTOLINEE has increased by +0.022, indicating a minor improvement in vegetation health. On the other hand, AUTOLINEE's NDVI shows a negative increase of −0.009, indicating a reduction in vegetation water content. In PIAZZA ILARIA ALPI, the NDVI shows a positive increment of +0.021, indicating a slight improvement in vegetation health. The NDWI of this site shows a negligible change of +0.009, suggesting a relatively stable water content. Both the NDVI and NDWI in VIA GOYA exhibit positive increments of +0.025 and −0.010, indicating a slight improvement in vegetation health but a reduction in water content. Finally, the NDVI reveals a minor change of −0.006 in VIA LEGNANO, indicating a reasonably steady plant development condition. The NDWI of this site showed a negative increment of −0.011. Further remote sensing investigations about the vegetation status of UPPER sites during post-interventions are programmed to be held in late 2023.

From 2022 to 2023, the NDVI index in city centers and individual areas, except for VIA LEGNANO, increased, suggesting an improvement in vegetation health and a positive impact of NbS interventions on increasing green cover. On the other side, the NDWI index showed a negative increment, except for PIAZZA ILARIA ALPI.

3.2.4. Considerations about NDVI and NDWI Variations (Baseline 2015–2022 and 2023)

The annual variations that occurred during the baseline time interval, 2015–2022, and 2023 in the NDVI and NDWI for each analyzed site are reported in Table 5.

By considering the indices averaged to the baseline 2015–2022 and the indices averaged for the values available for 2023, each analyzed area showed a positive increment of the NDVI and a negative increment of the NDWI. The NDVI and NDWI time series of 2015–2023 are reported in Figure 11.

**Table 5.** NDVI and NDWI variations (baseline 2015–2022 and 2023) for the city center of Latina and UPPER project sites.

| | **NDVI** | | | **NDWI** | | |
|---|---|---|---|---|---|---|
| | **2015–2022** | **2023** | **Increment** | **2015–2022** | **2023** | **Increment** |
| City center | 0.401 ± 0.163 | 0.472 ± 0.177 | +0.071 | −0.317 ± 0.131 | −0.362 ± 0.150 | −0.045 |
| CAMPO BOARIO | 0.373 ± 0.180 | 0.411 ± 0.198 | +0.038 | −0.262 ± 0.145 | −0.299 ± 0.125 | −0.038 |
| MERCATO | 0.308 ± 0.155 | 0.339 ± 0.183 | +0.030 | −0.229 ± 0.125 | −0.256 ± 0.144 | −0.026 |
| AUTOLINEE | 0.216 ± 0.157 | 0.253 ± 0.196 | +0.037 | −0.121 ± 0.142 | −0.165 ± 0.166 | −0.044 |
| PIAZZA ILARIA ALPI | 0.263 ± 0.165 | 0.322 ± 0.211 | +0.059 | −0.155 ± 0.143 | −0.200 ± 0.193 | −0.045 |
| VIA GOYA | 0.403 ± 0.143 | 0.442 ± 0.167 | +0.039 | −0.303 ± 0.116 | −0.336 ± 0.140 | −0.033 |
| VIA LEGNANO | 0.187 ± 0.131 | 0.201 ± 0.159 | +0.013 | −0.114 ± 0.116 | −0.165 ± 0.096 | −0.050 |

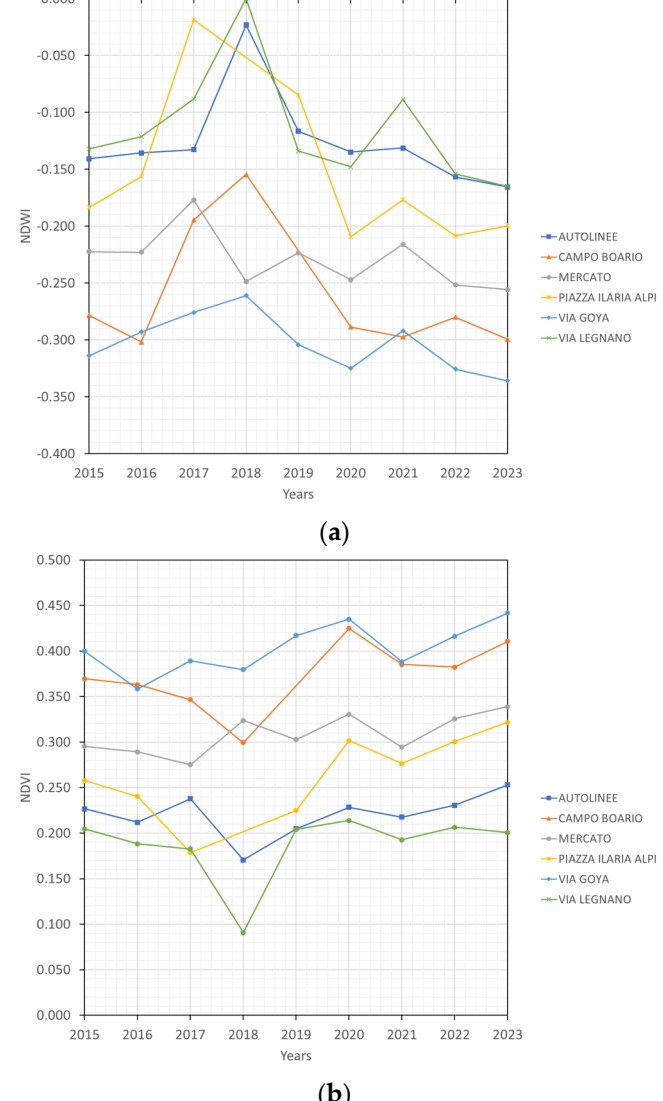

**(a)**

**(b)**

**Figure 11.** NDVI (**a**) and NDWI (**b**) time series of 2015–2023 for each analyzed UPPER site.

In order to compare the indices of the time interval 2015–2022 and the averaged indices of the years 2022 and 2023, ANOVA analyses were performed. The box plots of the mean NDVI and NDWI according to the considered time interval groups for each considered UPPER site are shown in Figures 12 and 13, respectively. The box plots of the NDVI and NDWI group means for the city center of Latina are shown in Figure 14.

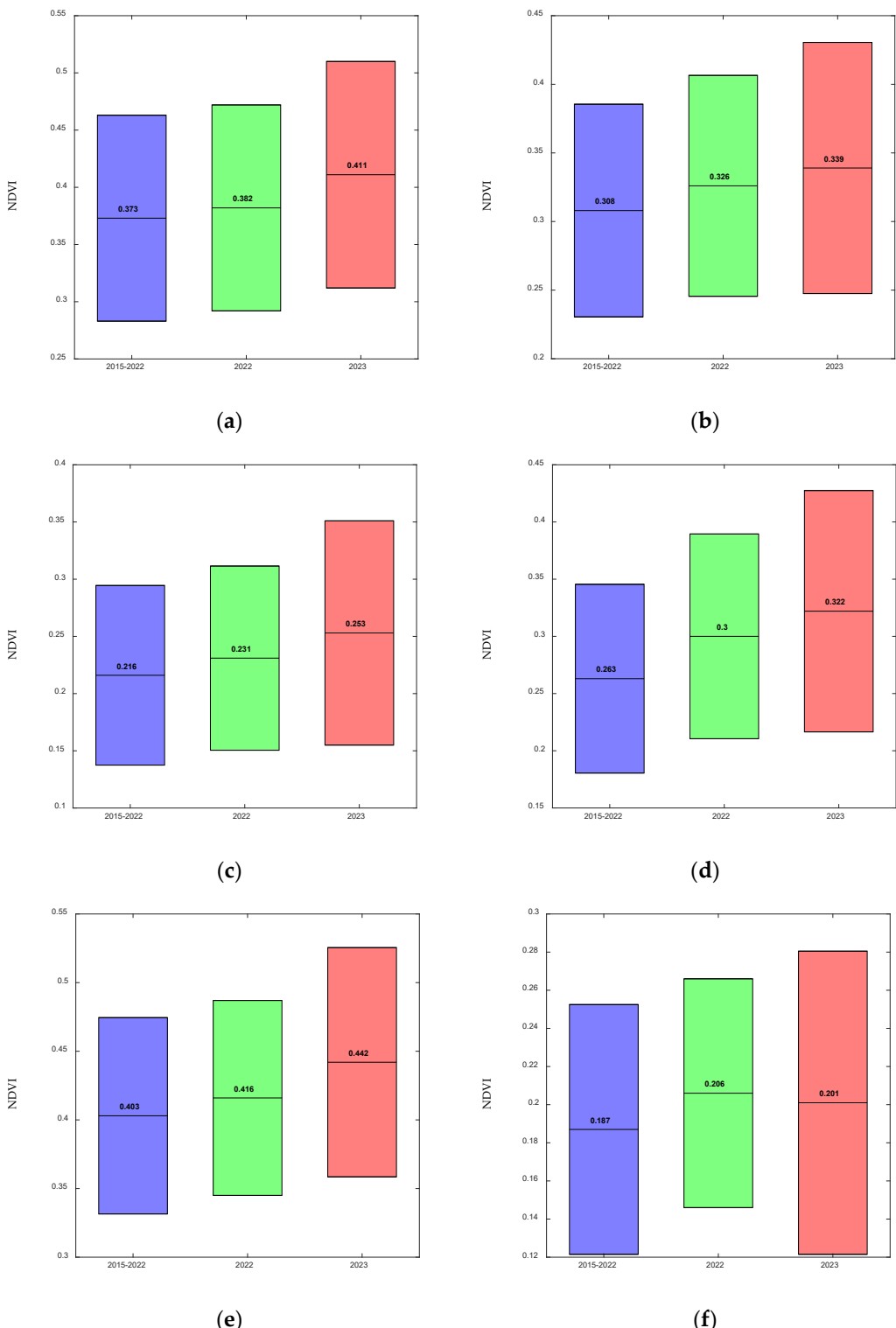

**Figure 12.** Box plots of NDVI group means with standard deviations for: CAMPO BOARIO (**a**), MERCATO (**b**), AUTOLINEE (**c**), PIAZZA ILARIA ALPI (**d**), VIA GOYA (**e**), and VIA LEGNANO (**f**).

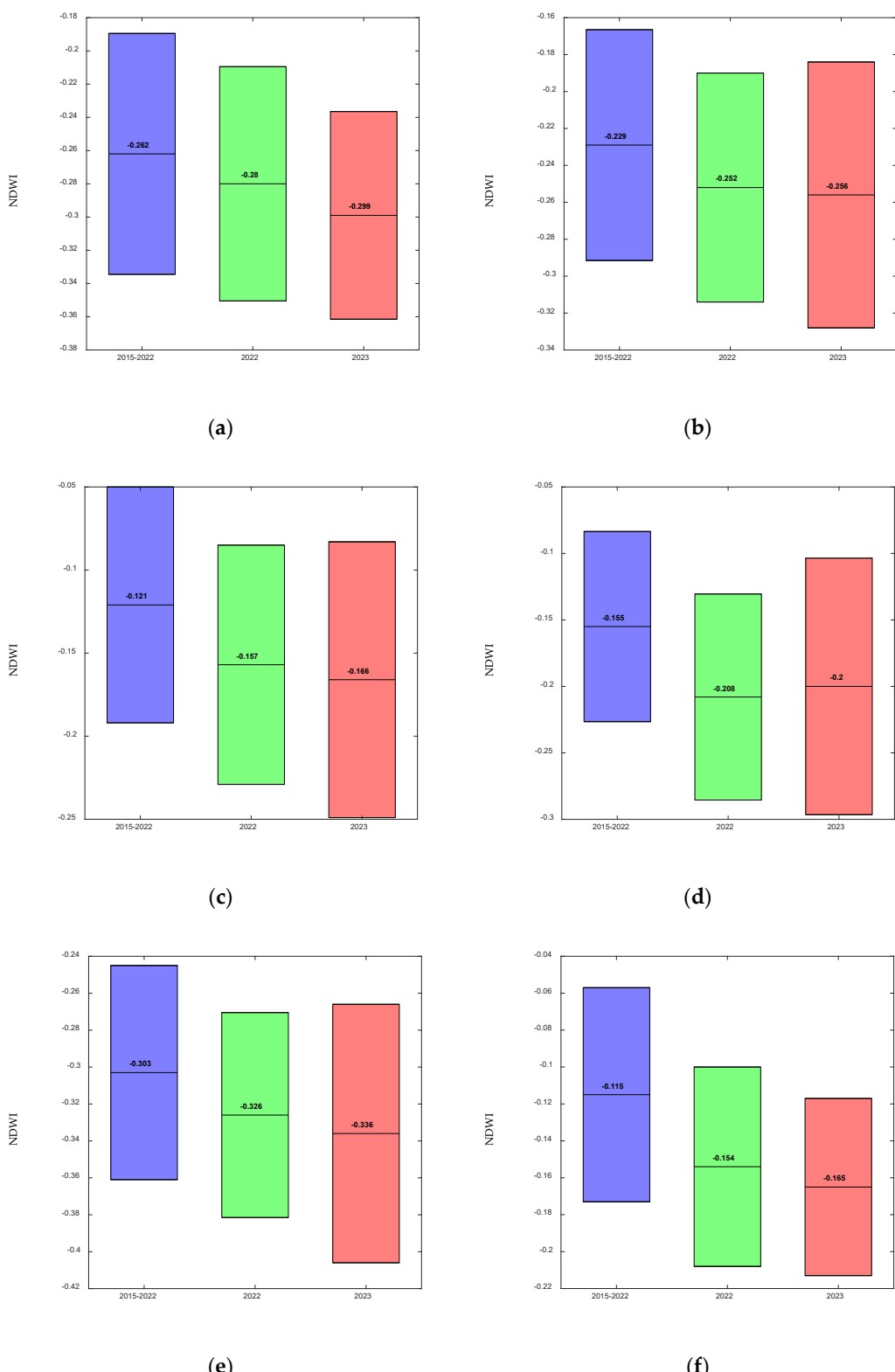

**Figure 13.** Box plots of NDWI group means with standard deviations for: CAMPO BOARIO (**a**), MERCATO (**b**), AUTOLINEE (**c**), PIAZZA ILARIA ALPI (**d**), VIA GOYA (**e**), and VIA LEGNANO (**f**).

According to the performed ANOVA analyses, for each analyzed site, there is no significant difference in NDVI and NDWI indices between the years 2015–2022, 2022, and 2023. Based on the available data, the variations observed of these indices are not statistically significant across the considered time intervals.

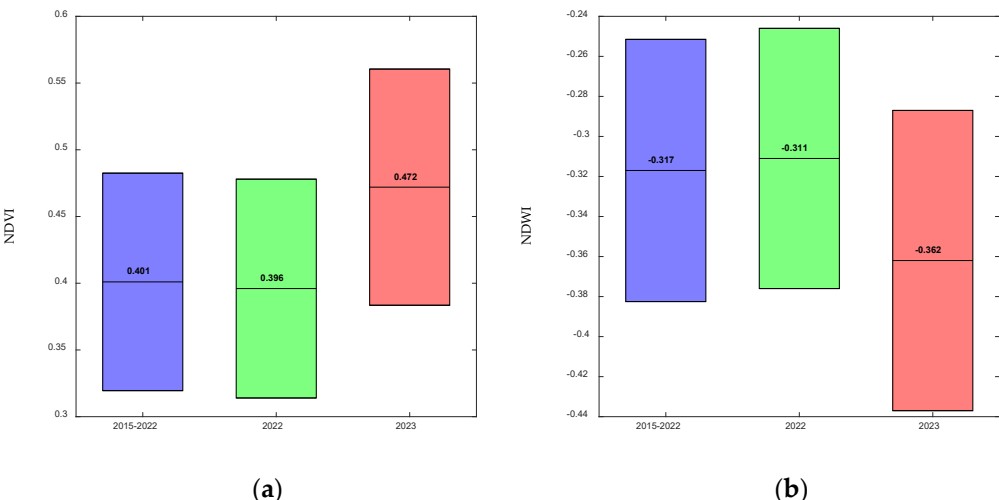

(**a**)　　　　　　　　　　　　　　　　　　　(**b**)

**Figure 14.** Box plots of NDVI (**a**) and NDWI (**b**) group means with standard deviations for the city center of Latina.

These results do not provide a comprehensive evaluation of NbS, but understanding the variations in these indices, they can provide valuable information for assessing water status and vegetation health in urban areas. This information can inform decision-making processes related to water management, landscape planning, and NbS implementation to enhance urban resilience and sustainability. Further research and analysis are needed to explore the implications of these indices' variations in relation to local environmental conditions, land management strategies, and long-term urban development goals.

Assessing the success and effectiveness of NbS requires considering a broader range of indicators, including ecosystem services, social benefits, and resilience to environmental challenges. Site-specific factors, management practices, and local environmental conditions should also be considered when interpreting the results. Further analysis and a holistic evaluation approach are needed to provide a comprehensive understanding of the performance and effectiveness of NbS in the studied areas.

### 3.3. Meteorological and Hydroclimate Data Analysis

A graphical representation of mean precipitation, temperature, and relative humidity for the time period 2015–2023 as evaluated from the meteorological station is reported in Figure 15.

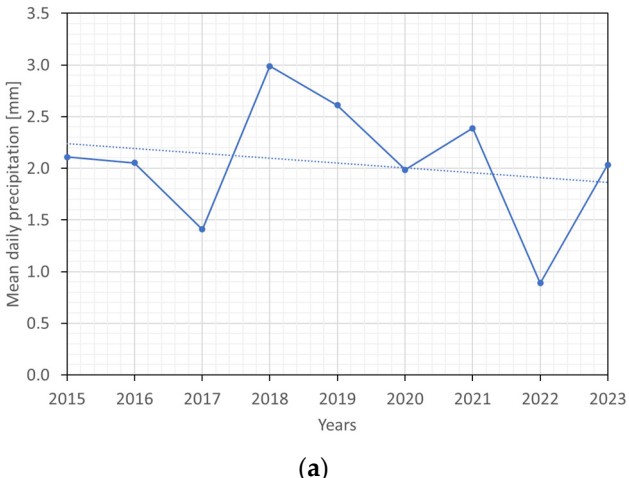

(**a**)

**Figure 15.** *Cont.*

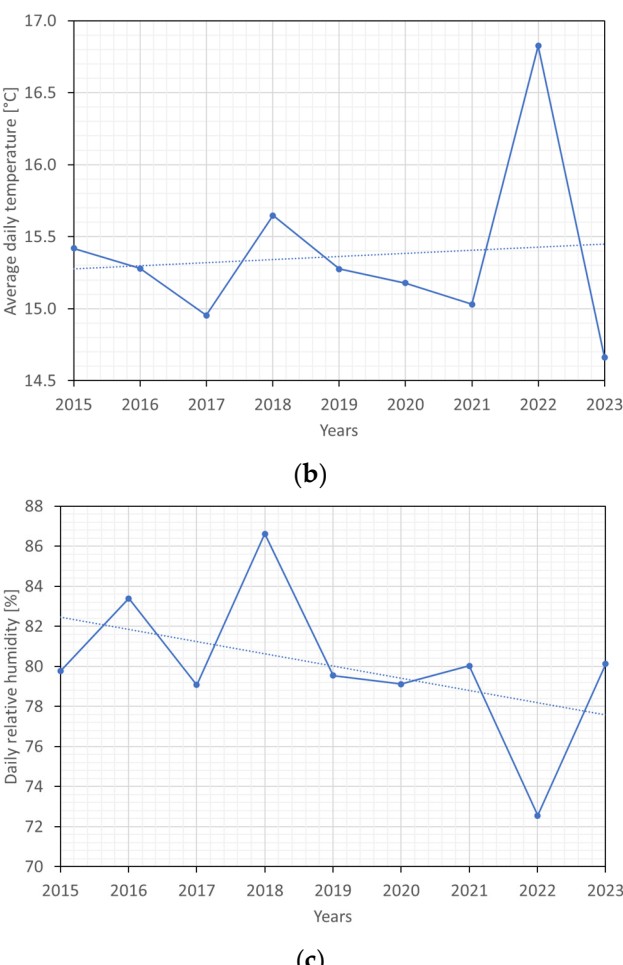

**Figure 15.** Mean daily precipitation (**a**), average daily temperature (**b**), and daily relative humidity (**c**), and their corresponding linear trend (dotted line).

As shown in Figure 15, the average daily precipitation and relative humidity exhibited a positive increasing trend, while the average daily temperature showed a positive rise. From the results obtained from PCA performed on the dataset containing the variables mean daily precipitation, maximum daily temperature, average daily temperature, minimum daily temperature, and daily relative humidity for the years 2015–2023, it can be observed that the year 2022 emerged as the hottest and driest. As shown from the PCA scores plot in Figure 16a, the score of the year 2022 is in the positive space of PC1, along with the year 2015, but is the most distant from the other scores. The variables that contribute to the positioning of the 2022 score are primarily those related to temperature, as can be seen in the loadings plot (Figure 16b).

However, based on the data recorded so far for the year 2023, it can be stated that, so far, it is a year with a milder climate compared to 2022, leaning more towards the values of the years prior to it.

In Figure 17 are reported the two drought-related variables SMI and SPI-3 for the time period 2015–2023 of the NUTS 3 IT44 Latina region. The SMI values remain within a narrow range during the analyzed time period, suggesting a consistent level of soil moisture across the years. On the other hand, most years show negative values of SPI-3, indicating drier conditions relative to the long-term average. The year 2023 exhibits a positive SPI-3 value, suggesting relatively wetter conditions compared to the preceding years.

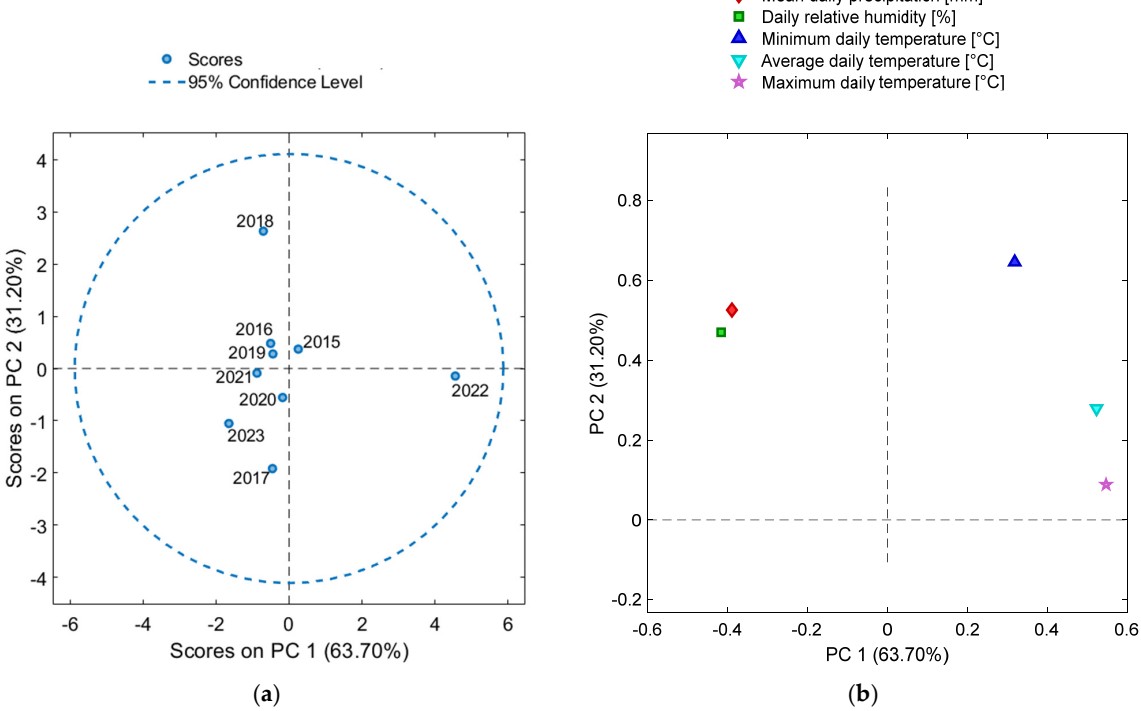

**Figure 16.** PCA scores plot (**a**) and loadings plot (**b**) of the first two PCs for the 2015–2023 meteorological data.

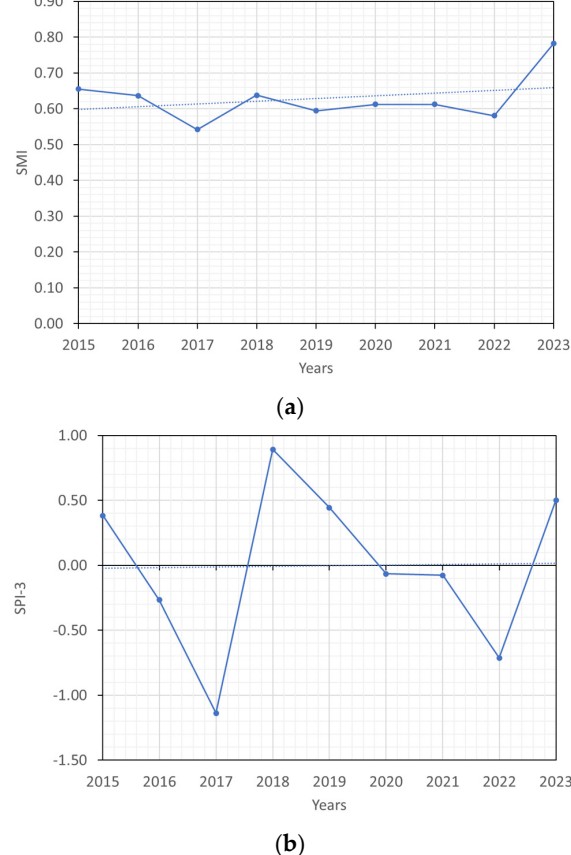

**Figure 17.** Soil Moisture Index (SMI) (**a**) and Standardized Precipitation Index for an accumulation period of 3 months (SPI-3) (**b**) for NUTS (nomenclature of territorial units for statistics) 3 level region IT44 Latina and their corresponding linear trends (dotted line).

From the results obtained from PCA performed on the dataset containing the variables mean daily precipitation, average daily temperature, daily relative humidity, SMI, and SPI-3 for the years 2015–2023, it can be observed that the years 2022 and 2017 are the most distant from years 2023 and 2018 in the space of the PC1 scores (Figure 18a). As can be seen from the loading plot, this fact is probably linked to the SMI values of 2023 (i.e., according to the collected data up until May 2023, the SMI value of this year has been one of the highest registered), while 2022 was strongly influenced by the high registered temperature, as also previously described using PCA, performed only on meteorological data.

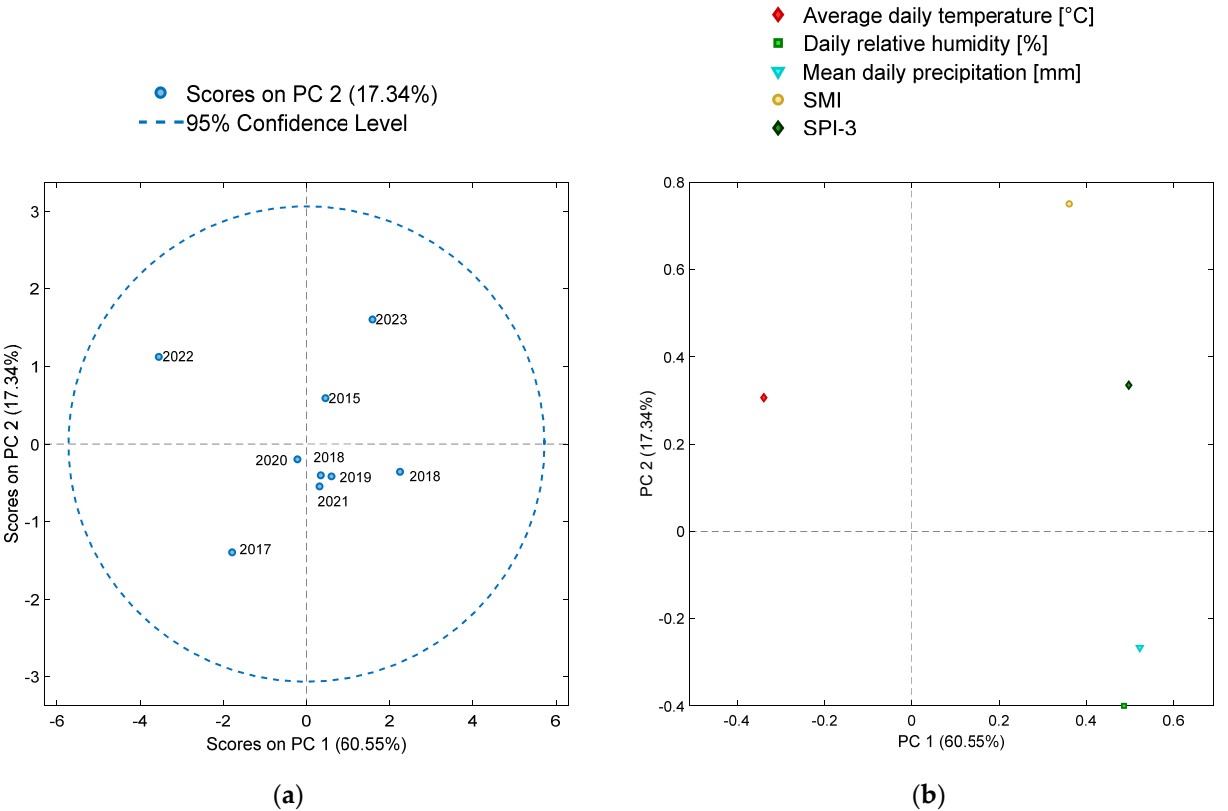

**Figure 18.** PCA scores plot (**a**) and loadings plot (**b**) of the first two PCs for the 2015–2023 meteorological and hydroclimatic data.

### 3.4. Correlation Analysis and PLS Regression Results

A correlation analysis was performed on the data spanning from 2015 to 2023 for the UPPER sites. The analysis included the variables average NDVI, average NDWI, mean daily precipitation, average daily temperature, daily relative humidity, SMI, and SPI-3.

Results, reported in Figure 19, show that there is a strong negative correlation ($\rho = -0.953$) between average NDVI and NDWI. A weak negative correlation also occurs for average NDVI and mean daily precipitation ($\rho = -0.147$) as well as between average NDVI and daily relative humidity ($\rho = -0.182$). The correlation between average NDVI and average daily temperature was found to be close to zero ($\rho = -0.012$), signifying a very weak relationship. Similarly, weak correlations were found between average NDWI and mean daily precipitation ($\rho = -0.013$), average NDWI and average daily temperature ($\rho = -0.090$), and average NDWI and daily relative humidity ($\rho = -0.037$). As expected, a strong positive correlation ($\rho = 0.839$) was found to occur between mean daily precipitation and relative humidity. Additionally, there was a moderate negative correlation between mean daily precipitation and temperature ($\rho = -0.424$), as well as between average daily temperature and relative humidity ($\rho = -0.475$). For how much correlation concerning drought indices, SMI showed a weak positive correlation with average NDVI mean ($\rho = 0.131$), mean daily

precipitation ($\rho = 0.290$), and daily relative humidity ($\rho = 0.268$). SPI-3 exhibited a positive correlation with SMI ($\rho = 0.647$) and mean daily precipitation ($\rho = 0.814$).

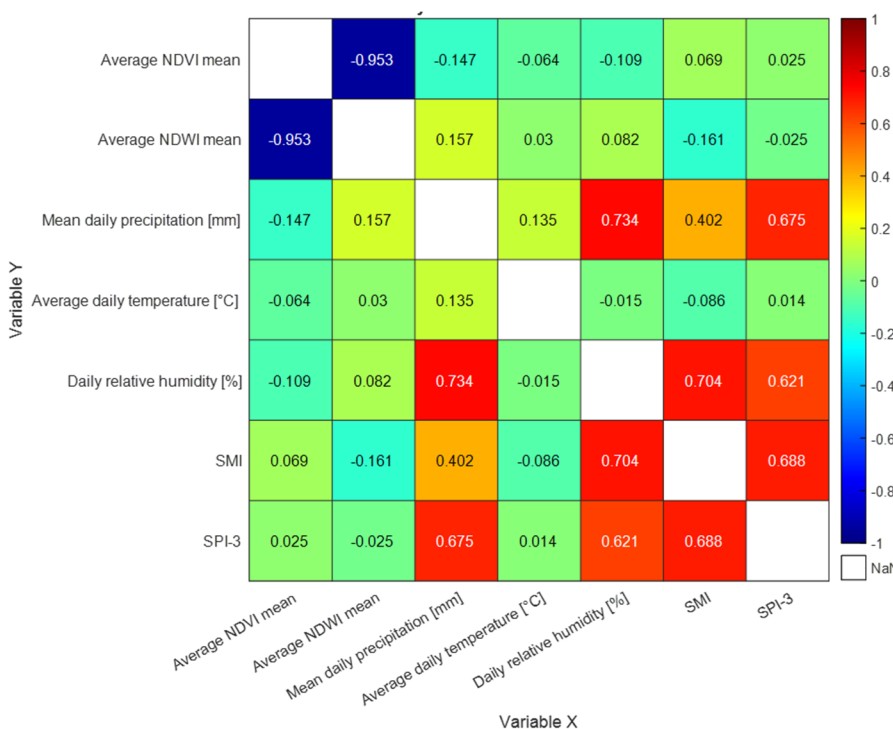

**Figure 19.** Heat map of the Spearman correlation coefficients of pairwise comparison between variables (Spearman's rank correlation matrix). The bar on the left side of the map shows the color of the Spearman correlation coefficients.

Each PLS regression was performed using two LVs. The statistical parameters of the performed PLS models are reported in Table 6.

**Table 6.** Statistical parameters of the PLS regression for UPPER project sites. RMSEC: Root Mean Square Error of Calibration; RMSECV: Root Mean Square Error of Cross-Validation; Bias (C): Bias of calibration; Bias (CV): Bias of Cross-Validation; $R^2_C$: coefficient of determination for calibration; $R^2_{CV}$: coefficient of determination for cross-validation.

| Site | AUTOLINEE | | CAMPO BOARIO | | MERCATO | | PIAZZA ILARIA ALPI | | VIA GOYA | | VIA LEGNANO | |
|---|---|---|---|---|---|---|---|---|---|---|---|---|
| Index | NDVI | NDWI | NDVI | NDWI | NDVI | NDWI | NDVI | NDWI | NDVI | NDWI | NDVI | NDWI |
| RMSEC | 0.004 | 0.017 | 0.023 | 0.029 | 0.014 | 0.012 | 0.026 | 0.032 | 0.018 | 0.012 | 0.015 | 0.023 |
| RMSECV | 0.013 | 0.046 | 0.068 | 0.105 | 0.024 | 0.027 | 0.067 | 0.101 | 0.043 | 0.032 | 0.063 | 0.088 |
| Bias (C) | 0.000 | 0.000 | 0.000 | 0.000 | 0.000 | 0.000 | 0.000 | 0.000 | 0.000 | 0.000 | 0.000 | 0.000 |
| Bias (CV) | 0.003 | −0.012 | 0.024 | −0.042 | 0.002 | −0.008 | 0.020 | −0.032 | 0.012 | −0.011 | 0.024 | −0.031 |
| $R^2_C$ | 0.962 | 0.816 | 0.588 | 0.708 | 0.476 | 0.726 | 0.594 | 0.760 | 0.375 | 0.666 | 0.846 | 0.763 |
| $R^2_{CV}$ | 0.638 | 0.033 | 0.134 | 0.402 | 0.038 | 0.000 | 0.335 | 0.244 | 0.085 | 0.000 | 0.003 | 0.039 |

Overall, the models seem to fit the calibration data well (indicated by low RMSEC and high $R^2_C$ values), but their performance on cross-validation was not as strong as in the calibration phase. Generally, the RMSEC values are much lower than the RMSECV values; this indicates that the model performs better on the data it was trained on compared to new data, suggesting a potential overfitting problem. For NDVI, $R^2_C$ values range from 0.375 to 0.962, indicating a moderate to strong correlation between the independent variables and the NDVI. For the NDVI, $R^2_C$ values range from 0.375 to 0.962, indicating a moderate

to strong correlation with the independent variables and NDVI. For the NDWI, the $R^2{}_C$ values are generally lower (0.666–0.816). Similarly, in cross-validation, the $R^2{}_{CV}$ for NDVI values range from 0.003 to 0.638, while values for NDWI range from 0 to 0.402.

The comparison between the satellite-evaluated indices and PLS-predicted values is shown in Table 7. The performed paired *t*-test of satellite-derived and PLS-predicted values for each index failed to reject the null hypothesis that there is no significant difference between the actual and predicted values. The obtained *p*-values (probability of obtaining the observed results if the null hypothesis is true) for the NDVI are 0.308 and 0.098 for the NDWI, respectively. Results obtained so far may indicate that the variations observed may not solely be attributable to climate fluctuations.

**Table 7.** Comparison between satellite-derived indices and PLS-predicted indices for each UPPER project site.

| Site | Year | Satellite Average NDVI | PLS Predicted NDVI | Satellite Average NDWI | PLS Predicted NDWI |
|---|---|---|---|---|---|
| AUTOLINEE | 2023 | 0.253 | 0.254 | −0.166 | −0.198 |
| CAMPO BOARIO | 2023 | 0.411 | 0.493 | −0.299 | −0.512 |
| MERCATO | 2023 | 0.339 | 0.311 | −0.256 | −0.243 |
| PIAZZA ILARIA ALPI | 2023 | 0.322 | 0.340 | −0.200 | −0.326 |
| VIA GOYA | 2023 | 0.442 | 0.416 | −0.336 | −0.321 |
| VIA LEGNANO | 2023 | 0.201 | 0.327 | −0.165 | −0.267 |

## 4. Conclusions

This research aimed to highlight the importance of monitoring and evaluating environmental performances of NbS interventions in urban contexts and showcases the role of proximal and remote sensing techniques in achieving comprehensive assessments.

As specifically seen for the UPPER project, by combining EO data and proximal sensing, a comprehensive and multi-scale monitoring approach for NbS can be achieved. NIRS provides rapid and cost-effective assessments of specific parameters, while remote sensing offers broader contextual information and detailed localized insights, respectively.

The results achieved by adopting proximal and remote sensing indices showed that both approaches allowed for correct area monitoring. The level of related environmental performance shows similar trends in all the ex-ante investigated areas. In any case, local-level vegetation health assessments using proximal sensing techniques will be essential to individually evaluate the status of the plants undergoing intervention. It is crucial to consider, in fact, that each plant species may have distinct requirements and tolerances, thus emphasizing the importance of considering the environmental context and plant-specific characteristics during the interpretation of future results at a larger scale.

This integrated approach significantly enhances our understanding of NbS' effectiveness, allowing for targeted interventions and evidence-based decision-making in the context of sustainable urban development. By combining these monitoring techniques, stakeholders can gain a holistic view of NbS' environmental impacts. This knowledge is essential for designing and implementing effective NbS strategies that address the challenges faced by urban environments.

As technology continues to advance, further improvements in EO data processing acquisition, in-field proximal sensing techniques, and data integration methodologies hold great promise for enhancing the assessment and management of NbS in urban settings. The integration of these monitoring approaches will contribute to a more sustainable and resilient future, where NbS play a vital role in addressing environmental challenges and promoting well-being in cities.

**Supplementary Materials:** The following supporting information can be downloaded at: https://www.mdpi.com/article/10.3390/su152216076/s1, Figure S1: Schematic representation of UPPER sites interventions for: VIA GOYA (a), PIAZZA ILARIA ALPI (b), VIA NEGHELLI (c), VIA LEPANTO (d), VIA ROMAGNOLI (e) and P.LE DEI MERCANTI. Figure S2: NDBI False color maps at 20 May 2022 (prior NbS interventions) and 5 May 2023: City center (a), CAMPO BOARIO (b), MERCATO (c), AUTOLINEE (d), PIAZZA ILARIA ALPI (e), VIA GOYA (f) and VIA LEGNANO (g). Figure S3: NDVI temporal series (2013–2023) of City center (a), CAMPO BOARIO (b), MERCATO (c), AUTOLINEE (d), PIAZZA ILARIA ALPI (e), VIA GOYA (f) and VIA LEGNANO (g). Figure S4: NDWI temporal series (2013-2023) of City center (a), CAMPO BOARIO (b), MERCATO (c), AUTOLINEE (d), PIAZZA ILARIA ALPI (e), VIA GOYA (f) and VIA LEGNANO (g).

**Author Contributions:** G.B.: Conceptualization, Methodology, Resources, Writing—Review and Editing, Visualization, Supervision. R.G.: Conceptualization, Methodology, Software, Validation, Formal Analysis, Investigation, Data Curation, Writing—Original Draft, Writing—Review and Editing, Visualization. S.S.: Validation, Resources, Writing—Review and Editing, Visualization. All authors have read and agreed to the published version of the manuscript.

**Funding:** This study was carried out within the Urban Productive Parks for the Development of NbS Related Technologies and Services (project number: UIA04-252; CUP: B23C19000020002; Main Urban Authority Municipality of Latina) supported by Urban Innovative Actions (UIA)—Initiative of the European Union promoting pilot projects in the field of sustainable urban development.

**Institutional Review Board Statement:** Not applicable.

**Informed Consent Statement:** Not applicable.

**Data Availability Statement:** The data presented in this study are available from the corresponding authors upon reasonable request.

**Conflicts of Interest:** The authors declare no conflict of interest.

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
