# Peer review of "Assessing the Environmental Performances of Nature-Based Solutions Implementation in Urban Environments through Visible and Near-Infrared Spectroscopy: A Combined Approach of Proximal and Remote Sensing for Monitoring and Evaluation"

_sustainability, doi:10.3390/su152216076_

Round 1

Reviewer 1 Report

Comments and Suggestions for Authors

Thank you very much for very interesting and important work. It provides good review on Nature-Based Solutions (NbS) and their importance in  environmental challenges and sustainable development.

It is a very good and interesting approach like combining of remote sensing and proximal sensing techniques for the environmental monitoring of NbS interventions.

I found the manuscript written well and it provides good understanding of NbS.

There are also good work and analyses presented regarding to proximal and remote sensing.

Having said all above comments I still miss an important analyses and conclusions which I believe it must be included in the manuscript.

This missing part has already been addressed in the manuscript as well like 

"In each case study, it is crucial to verify the impacts of NbS through monitoring and evaluation processes. (Page 4, Line 150)"

There are changes presented from 2022 to 2023. But what is important that what are interventions of NbS for each case study. We see changes but we do not know link between these interventions and their effect and also analyses. 

Another important point that there are about 10 years Sentinel-2 analysed. It would be interesting to see changes without interventions of NbS and then we can better understand the meaning of changes with interventions of NbS.

Author Response

[Sustainability] Manuscript ID: sustainability-2548641

The authors wish to thank the Editor and the Reviewers for the revisions and the suggestions.

The requested answers and the integrations are reported In the following attachment. The revised text is reported with the “Track Changes” function in the manuscript, according to Reviewers requests/suggestions.

Reviewer 2 Report

Comments and Suggestions for Authors

Review report on the ms entitled “Assessing the environmental performances of Nature-Based Solutions …” by Giuseppe Bonifazi et al.

The ms focuses on the analysis of the welfare of vegetation in an urban environment and quantitative assessment of the effect of a local project aiming at the its improvement. The ms fits in the scope of the journal, employs some established remote sensing methodology, and thus could be potentially considered for publication. However, certain questions arise while reading the ms, that may require additional clarification:

1. The Introduction is too long, coming to the problem from somewhere very afar, with long phrases likely repeating statements from earlier works and recommendation documents. As nowadays these documents are typically available online, there is no need for such as excessive quotation, as well as such a long and unspecific discussion. The problem of monitoring and evaluation could be highlighted, and limitations of existing methods could be highlighted outright.

2. The authors claim comparison between sets of similar measurements performed in May 2022 and May 2023. But each year is unique in terms of phenological characteristics, depending on many variables (climate variations, meteorological conditions prior and during the measurements, from long-term to very short-term effects). Did the authors take into account associated hydroclimatic and meteorological information? Some basic indices such as temperature and precipitation in the short-term, as well as some measure of water content (such as Palmer Drought Severety Index or another similar metric) should be compared.

3. The authors claim to have access to the data between 2013 and 2023 from the same source. Taking this into account, it would be reasonable to proof that neither the year 2022 nor the year 2023 are exceptional from the typical conditions. Possible outliers such as seasonal creep effects could affect the results. More specifically, a solid proof would require statistical assessment of the 2022 and 2023 records against baseline 2013-2022 baseline. Only if the 2022 results are within, and 2023 results are outside the confidence interval for 2013-2022, one could refer to the results as an effect of the project.

4. Vegetation dynamics follows hydroclimatic variations, that are known to exhibit long-term persistence. This drastically affects the confidence intervals of the respective estimates (see, e.g., https://doi.org/10.1073/pnas.1700838114 and references therein). In turn, respective corrections are required to correctly evaluate the results, and to prove that the observed discrepancies between 2022 and 2023 are neither an artifact of long-term variability of the studied vegetation metrics nor an effect of a longer trend in the hydroclimatic variability, but indeed the outcome of the current project in a statistically valid manner.

5. The results should be rather presented in a more systematic manner. Boxplots with highlighted statistically significant discrepancies (with reference to methods, confidence intervals/probabilities, as well as preconditions for those, such as normality, whereever it is required by the method), are essential.

In summary, the ms requires a revision towards a more concise description of the results, and especially their thorough statistical validation following accurate methods that take into account possible contributions from random (although correlated) hydroclimatic variations. In recent years, significant methodological research in this direction led to the enhancement of the implied methodology, including rethinking of dendrological data from the past and understanding current vegetation dynamics. This ms seems to lack these essential validations, preventing from its recommendation for publication in its current form.

Author Response

(The authors gave the same response as above.)

Round 2

Reviewer 2 Report

Comments and Suggestions for Authors

The manuscript has improved considerably since the initial evaluation. The authors performed additional comparisons against earlier vegetation dynamics since 2015 and provided results of statistical analysis using PCA, ANOVA, and pairwise statistical tests. Altogether, the results sound reasonable and confirm certain doubts regarding the statistical significance of the discrepancies between vegetation indices on specific years that the authors now confirmed directly on page 26, lines 638–639. Moreover, newly provided information on the local climate variations also indicates that the year 2022 was rather anomalous, and thus the positive dynamics in the vegetation indices in the year 2023 compared to the year 2022 could also be contributed by more favourable climate conditions compared to the extremely hot and dry year 2022.

However, there are some methodological issues and technical remarks that remain in connection with the results of this study and their presentation that could be taken into account prior to publication:

1) The authors refer to the mean daily precipitation, average daily temperature, and daily relative humidity, but provide a single point per year. Does this mean that the annual mean values have been taken into account in the analysis? While it is more common to refer to drought severity indices, such as the Palmer index, in vegetation dynamics studies, the approach by the authors may still be legitimate if the multispectral observations have also been collected several times per year during different seasons. Then, since both quantities are averaged over the entire year, on average, there is not much difference between the analysis of the amount of precipitation directly or a more sophisticated quantity such as PDSI, which also takes into account the accumulation of water content in the soil, etc.

2) Figs. 12–14 show boxplots for 2015–2022, as well as 2022 and 2023 as individual years. Quite surprisingly, they typically exhibit similar standard deviations. If the data in the first box were averaged over several years, the standard deviation should have been reduced accordingly, a well-known effect in statistical analysis. This effect seems to be entirely missing from all these plots, which raises questions about how these data were calculated. Did the authors somehow calculate the respective data for each year and then average the standard deviations over all years from 2015 to 2022? If you just average the data, the expected outcome is very different, with considerably smaller standard deviations.

3) Figs. 12–14 have “means” as the Y-label, which is very unspecific. I would rather advise displaying the underlying quantity for immediate transparency for the reader and referring to the fact that the respective means of this quantity over certain periods have been displayed in the figure caption only.

4) In Section 3.4, the authors refer to the correlation coefficients between vegetation indices and hydroclimatic variables. While correlation coefficients are indeed the basic quantities, a more correct approach to the evaluation of the relative contributions could be performed by considering partial correlations. Since the authors combine PCA with correlation analysis and also use Matlab, in the opinion of this reviewer, a more straightforward and also interpretable approach would be to consider a partial correlation-based model, such as a Bayesian network. There are existing tools for that already implemented in Matlab with a variety of application examples from climate, hydrology, and meteorology that could even be used as prototypes; see, e.g., https://doi.org/10.1016/j.softx.2020.100588. The approach could be as simple as:

-- In the first (training) mode, the model could be trained for the 2015–2022 period, with climate variability indicators as input variables and vegetation dynamics as output variables. At this step, the model is pre-trained based on the partial correlations between the respective variables.

-- In the second (inference) mode, the analysis is extended to the year 2023 by providing climate indicators for the year 2023 as input variables and predicting the vegetation indices for the year 2023 solely from the model.

-- In the third (validation) mode, the predicted vegetation indices are compared against their actual observations during the year 2023. If there are statistically significant improvements (in terms of statistical tests, that could be either a T-test in the case of Gaussian or a U-test in the case of non-Gaussian data distributions), this would directly confirm that: (a) the vegetation improved in the year 2023 following the intervention, and (b) this improvement in fact appeared beyond that which could be expected solely due to the more favourable hydroclimate conditions of the year 2023 compared to the year 2022.

Indeed, one does not need to focus on the specific tool quoted above; a qualitatively similar statistical analysis design could also be implemented by combining PCA, correlation analysis, and statistical tests as well. The above example of a stepwise procedure is only to highlight how the outcome of the statistical analysis could be presented in a more systematic and transparent manner for the reader and clearly address the issue of the evaluation of the specific impact of the intervention while eliminating the contributions of climate variations from the analysis.

Round 3

Reviewer 2 Report

Comments and Suggestions for Authors

The ms has been further improved in several directions and particularly in terms of the statistical analysis and interpretation. Specifically, when comparing the empirical and predicted NDWI index values, the authors find p < 0.1. Although this remains above the standard p = 0.05 threshold, this nevertheless indicates significant changes in the water content at ~90% confidence level, which is a reasonable finding. Therefore, I am happy to recommend the revised ms for publication in its present form.